# Combinatorial expression of neurexins and LAR-type phosphotyrosine phosphatase receptors instructs assembly of a cerebellar circuit

Alessandra Sclip [1] ✉ & Thomas C. Südhof [1,2] ✉

Synaptic adhesion molecules (SAMs) shape the structural and functional properties of synapses and thereby control the information processing power of neural circuits. SAMs are broadly expressed in the brain, suggesting that they may instruct synapse formation and specification via a combinatorial logic. Here, we generate sextuple conditional knockout mice targeting all members of the two major families of presynaptic SAMs, Neurexins and leukocyte common antigen-related-type receptor phospho-tyrosine phosphatases (LAR-PTPRs), which together account for the majority of known trans-synaptic complexes. Using synapses formed by cerebellar Purkinje cells onto deep cerebellar nuclei as a model system, we confirm that Neurexins and LAR-PTPRs themselves are not essential for synapse assembly. The combinatorial deletion of both neurexins and LAR-PTPRs, however, decreases Purkinje-cell synapses on deep cerebellar nuclei, the major output pathway of cerebellar circuits. Consistent with this finding, combined but not separate deletions of neurexins and LAR-PTPRs impair motor behaviors. Thus, Neurexins and LAR-PTPRs are together required for the assembly of a functional cerebellar circuit.

In neural circuits, synapses operate as the fundamental computational units that process a presynaptic signal into a postsynaptic response. Synapses are thought to be organized by synaptic adhesion molecules (SAMs) that direct the assembly and specification of synaptic junctions in the brain, enable synaptic plasticity, and regulate synapse stability and *turnover*[1–5]. Many SAMs likely cooperate in organizing synapses. Among these, two families of evolutionarily conserved presynaptic adhesion molecules stand out: Neurexins and leukocyte common antigen-related -type receptor phospho-tyrosine phosphatases (LAR-PTPRs, a.k.a. LAR-RPTPs) (Fig. 1a)[6–10]. Neurexins and LAR-PTPRs are remarkable because they bind to a large number of postsynaptic adhesion molecules and form multifarious trans-synaptic complexes that together account for the majority of all known trans-synaptic interactions.

Neurexins and LAR-PTPRs are each encoded by three genes (*Nrxn1-3* and *Ptprd*, *Ptprf*, and *Ptprs* in mice; here, we refer to the latter as PtprD, PtprF, and PtprS to render the isoform names more recognizable). The primary transcripts of *Nrxns* and LAR-PTPR genes are extensively alternatively spliced, creating thousands of isoforms[11–16]. Moreover, all neurexin genes contain multiple promoters directing transcription of longer α- and shorter β-neurexins[17–20], and the *Nrxn1* gene contains a further promoter for γ-neurexin isoforms[21]. All neurons co-express neurexins and LAR-PTPRs[12,22,23], suggesting that these SAMs represent fundamental building blocks of synaptic circuits. Neurexins and LAR-RPTPs share no sequence similarity and largely bind to different postsynaptic adhesion molecules. Only one postsynaptic ligand, Neuroligin-3, is known to interact with both neurexins and a LAR-PTPR[24]. In addition, neurexins and LAR-PTPRs bind to each

[1]Department of Cellular and Molecular Physiology, Stanford University School of Medicine, Stanford, CA 94305, USA. [2]Howard Hughes Medical Institute, Stanford University School of Medicine, Stanford, CA 94305, USA. ✉e-mail: sclip.alessandra@gmail.com; tcs1@stanford.edu

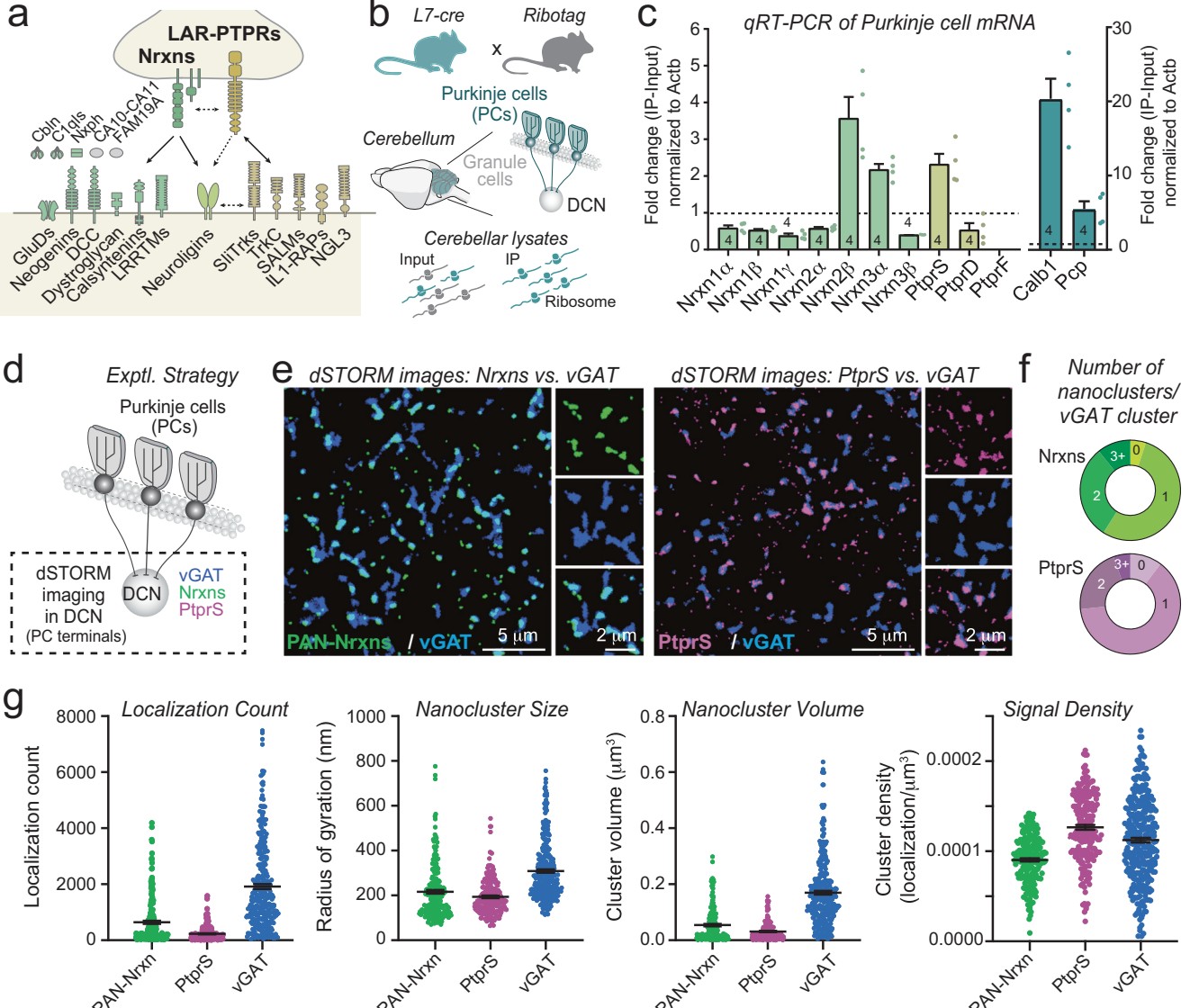

**Fig. 1 | Nanoclusters of Neurexins and PtprS are localized in the same inhibitory synapses of deep cerebellar nuclei in mice. a** Schematic of selected trans-synaptic adhesion complexes formed by Neurexins and LAR-PTPRs. **b**, **c**, Profiling of Neurexin and LAR-PTPR transcripts in cerebellar Purkinje cells. Experimental design (**b**) and quantification (**c**; means ± SEM; $n = 4$ mice (2 females and 2 males)) of ribosome-bound mRNAs (IP) from the cerebellum of RiboTag mice crossed to L7-cre mice (which express Cre only in Purkinje cells) followed by quantitative RT-PCR. Data were normalized to actin and compared to total mRNA (input). PtprF mRNA was not detected in the ribosome-bound fraction. Calb1 and Pcp were used as specific markers for Purkinje cells. **d**–**g** Localization of both Neurexins and PTPRS in most synapses of deep cerebellar nuclei (DCN) that receive inputs from Purkinje

cells. **d** Experimental design for **e**–**g**. **e** Representative images for Neurexin (green, 647) and PTPR clusters (magenta, 647) visualized using direct stochastic optical reconstruction microscopy (dSTORM); Purkinje cell boutons in the DCN are visualized with antibodies to vGAT (blue, 568); **f**, pie charts showing the distribution of Purkinje cells boutons without (0) or with Neurexins (green) or PTPRS (magenta) clusters (1, 2, and 3+); **g**, violin plots illustrating the properties of Neurexins (green), PTPRS (magenta) and vGAT (blue) nanoclusters as analyzed by dSTORM; $n = 213$ (5 ROIs/ 1 mouse) clusters for Nrxn, $n = 204$ (5 ROIs/ 1 mouse) clusters for PTPRS, $n = 317$ (10 ROIs/ 1 mouse) clusters for vGAT. Source data are provided within the Source Data file.

other in cis via the neurexin carbohydrate modifications[25,26], suggesting a possible functional relationship between neurexins and LAR-PTPRs.

Extensive studies demonstrated a central role for neurexins and LAR-PTPRs in shaping synapse properties, but puzzlingly neither neurexins nor LAR-PTPRs appear to be generally required for synapses as such[27–33]. The lack of a requirement of neither neurexins nor LAR-PTPRs for synapse formation was unexpected given that these molecules exhibit strong synaptogenic activities in heterologous synapse formation assays (reviewed here[8,9,34]). A possible explanation for the lack of a requirement of neither neurexins nor LAR-PTPRs for synapse formation is that they are functionally redundant. To test this

hypothesis and to explore the possibility that combinatorial expression of neurexins and LAR-PTPRs could encode the formation and organization of synapses, we here generate sextuple conditional knockout (6cKO) mice targeting all isoforms of neurexins (except for Nrxn1γ) and all isoforms of LAR-PTPRs. We analyze the sextuple neurexin and LAR-PTPRs deletions and compare them to triple deletions of neurexins or LAR-PTPRs, using synapses formed by cerebellar Purkinje cells onto the deep cerebellar nuclei (PC→DCN synapses) as a model system. Our data reveal that the sextuple conditional deletion of neurexins and LAR-PTPRs largely ablates PC→DCN synaptic connections as monitored by super-resolution imaging, pseudo-rabies virus tracing, and electrophysiology, whereas the separate triple

neurexin or LAR-PTPR deletions do not. Thus, neurexins and LAR-PTPRs are functionally redundant in PC→DCN synapses, suggesting that these synapses are assembled via combinatorial expression of multiple trans-synaptic complexes.

## Results

To test the possibility that neurexins and LAR-PTPRs might be functionally redundant in synapse formation, we aimed to identify a central synapse that contains neurexins and LAR-PTPRs and that is amenable to functional analyses. A survey of single-cell RNAseq data showed that cerebellar Purkinje cells co-express high levels of both neurexins and LAR-PTPRs (Figure S1a). RiboTag profiling of Purkinje cell mRNAs by quantitative RT-PCR revealed that expression of at least two neurexins, Nrxn2β and Nrxn3α, and of PtprS is enriched in Purkinje cells (Fig. 1b, c). Furthermore, single-cell expression data showed that neurons of the deep cerebellar nuclei, the major synaptic targets of Purkinje cells[35–37], co-express many postsynaptic ligands of neurexins and LAR-PTPRs (Figure S1b), suggesting that Purkinje cell synapses formed on deep cerebellar nuclei (PC→DCN synapses) may be well-suited for analysis of the redundancy of neurexins and LAR-PTPRs.

The cerebellar cortex consists of a relatively simple, stereotyped circuitry that is assembled postnatally, making it an ideal system to study how neuronal connections are formed. The only output pathway from the cerebellar cortex is provided by Purkinje cells (PCs) that are characterized by a large dendritic tree and a single axon that forms an inhibitory projection to the deep cerebellar nuclei (DCN). The DCN forms the only output pathway of the cerebellum; PC→DCN synapses are thus placed in a central position in the cerebellar circuit[35–37]. To examine whether PC→DCN synapses express both neurexins and LAR-PtprS, we labeled cryosections of deep cerebellar nuclei with antibodies to PtprS or to neurexins (Fig. 1d–g). The neurexin antibodies were raised against the C-terminus of Nrxn1 but appear to cross-react with all neurexins[38]. We co-stained the sections with antibodies to vGAT as a marker of PC→DCN inhibitory synapses to identify synaptic junctions and analyzed the sections by dSTORM (Fig. 1d, S2).

The resulting images revealed that neurexins and PtprS are both present in synapses and that they localize in synapses to one or more 'nanoclusters' (Fig. 1e, f). Most vGAT-positive synaptic junctions (>85%) contained at least one neurexin and PtprS nanocluster. Approximately 25–33% of synapses featured more than one nanocluster (Fig. 1f). The neurexin and PtprS nanoclusters were similar in size and volume (Fig. 1g), resembling nanoclusters previously characterized for neurexins and other SAMs in multiple studies[38–44]. Thus, neurexins and at least one LAR-PTPR isoform, PtprS, are co-localized in most PC→DCN synapses and are parts of nanoclusters.

We next tested the potential redundancy among neurexins and LAR-PTPRs at PC→DCN synapses. We crossed triple neurexin conditional KO mice (Nrxn123 3cKO mice[30]) and triple LAR-PTPR conditional KO mice (PtprDFS 3cKO mice[33]) with PV-Cre driver mice and with each other to generate triple Nrxn123 3cKO and PtprDFS 3cKO mice and sextuple Nrxn123-PtprDFS 6cKO mice containing or lacking a PV-Cre allele (Fig. 2a). In these mice, PV-Cre drives deletion of neurexins and/or LAR-PTPRs in parvalbumin-positive neurons such as Purkinje cells and molecular layer interneurons in the cerebellum as well as multiple other types of neurons in the brain. PV-Cre thus mediates the deletion of neurexins and LAR-PTPRs in Purkinje cells, enabling us to study the effect of such deletions on PC→DCN synapses.

All neurexin and LAR-PTPR mutant mice were viable and fertile. However, the PV-Cre Nrxn123-PtprDFS 6cKO mice exhibited major motor behavior abnormalities. Given the widespread expression of parvalbumin in the brain, the phenotypes of the triple and sextuple Nrxn123 and PtprDFS cKO/PV-Cre mice could be due to a synaptic dysfunction in multiple brain regions, including the cerebellum. Nevertheless, the relative effects of the triple vs. the sextuple deletions were highly informative. Measurements of the spontaneous

movements of mice in an open field revealed no changes in the distance traveled or time in the center of the field in Nrxn123 3cKO PV-Cre or in PtprDFS 3cKO PV-Cre mice, but a significant increase in the distance traveled in Nrxn123/PtprDFS 6cKO mice (Fig. 2b, c). Moreover, Nrxn123 3cKO PV-Cre exhibited tremor that was severely exacerbated in Nrxn123-PtprDFS 6cKO mice but absent from PtprDFS 3cKO PV-Cre mice (Figs. 2c, S3a). These results are consistent with previous data showing that dysfunction of deep cerebellar nuclei causes an essential tremor-like syndrome[45].

To further analyze motor behaviors, we examined the gait of mice (Fig. 2d, e, S3b). Nrxn123 3cKO PV-Cre and PtprDFS 3cKO PV-Cre mice exhibited only minor changes in gait properties, whereas Nrxn123-PtprDFS 6cKO mice suffered from a major impairment that manifested as a near doubling in the paw overlap and a major increase in paw based (Fig. 2e). Next, we analyzed the mice on an accelerating rotarod (4–40 rpms, maximum 5 min) over three days with three trials a day. Both the initial coordination and the rate of learning on the accelerating rotarod were decreased approximately two-fold in Nrxn123 3cKO PV-Cre compared to littermates, whereas PtprDFS 3cKO PV-Cre mice had no phenotype (Fig. 2f, g). Nrxn123-PtprDFS 6cKO mice, however, presented with a massively aggravated phenotype as they were unable to master the rotarod task at all and fell off the accelerating rod immediately (Fig. 2f, g).

Together these data show that the Nrxn123 3cKO PV-Cre mice but not the PtprDFS 3cKO PV-Cre mice exhibit a significant motor behavior impairment that is greatly enhanced in Nrxn123-PtprDFS 6cKO mice, consistent with an important role especially for neurexins in synaptic transmission and with the notion that neurexins and LAR-PTPRs may be functionally redundant. These phenotypes were, at least in part, likely due to cerebellar dysfunction given that motor behaviors are dependent on the cerebellum. Here, the phenotypes were not developmentally conditioned since morphological analyses of the cerebellum of Nrxn123-PtprDFS 6cKO mice showed that the volume and overall cytoarchitecture of the cerebellum (Fig. 3a–h, Figure S4), as well as the targeting of Purkinje cell axons to deep cerebellar nuclei (Figure S5),S5) were not significantly altered.

To determine whether the number of PC→DCN synapses as the major cerebellar output pathway was affected by the neurexin and/or LAR-PTPR deletions, we analyzed these synapses by immunocytochemistry for vGAT and calbindin in cryosections (Fig. 4a, b). As analyzed by confocal microscopy, neither Nrxn123 3cKO PV-Cre nor PtprDFS 3cKO PV-Cre mice exhibited a significant decrease in average vGAT staining intensity or in the vGAT-stained area as proxies for synapse numbers (Fig. 4b). However, the PtprDFS 3cKO mice but not the Nrxn123 3cKO mice displayed a small right-shift in the cumulative distribution of the staining intensity and stained area (Fig. 4b). These results suggest that the triple Nrxn123 and PtprDFS deletions do not decrease PC→DCN synapse numbers, but in the case of the triple PtprDFS deletion may even cause a small increase. In contrast to the triple deletions, the sextuple Nrxn123-PtprDFS 6cKO deletion induced a large decrease in vGAT staining intensity and stained area, with the latter nearly halved (Fig. 4b).

The imaging result is indicative of a synapse loss in Nrxn123-PtprDFS 6cKO mice, but confocal microscopy is unable to resolve individual synapses. To confirm synapse loss, we examined synapse density in cryosections of the deep cerebellar nuclei of littermate Nrxn123/PtprDFS 6cKO control and PV-Cre mice by dSTORM after double-labeling synapses with antibodies to vGAT as a marker of inhibitory synaptic vesicles and Munc13 as a marker of presynaptic active zones (Fig. 4c). Synaptic vGAT and of Munc13 clusters were both reduced in density approximately two-fold upon deletion of neurexins and LAR-PTPRs, whereas the volumes and other properties of these clusters were not significantly altered (Figs. 4d, e, S6a–f). Thus, the deletion of all major neurexin and LAR-PTPR isoforms from Purkinje cells causes a loss of approximately half of

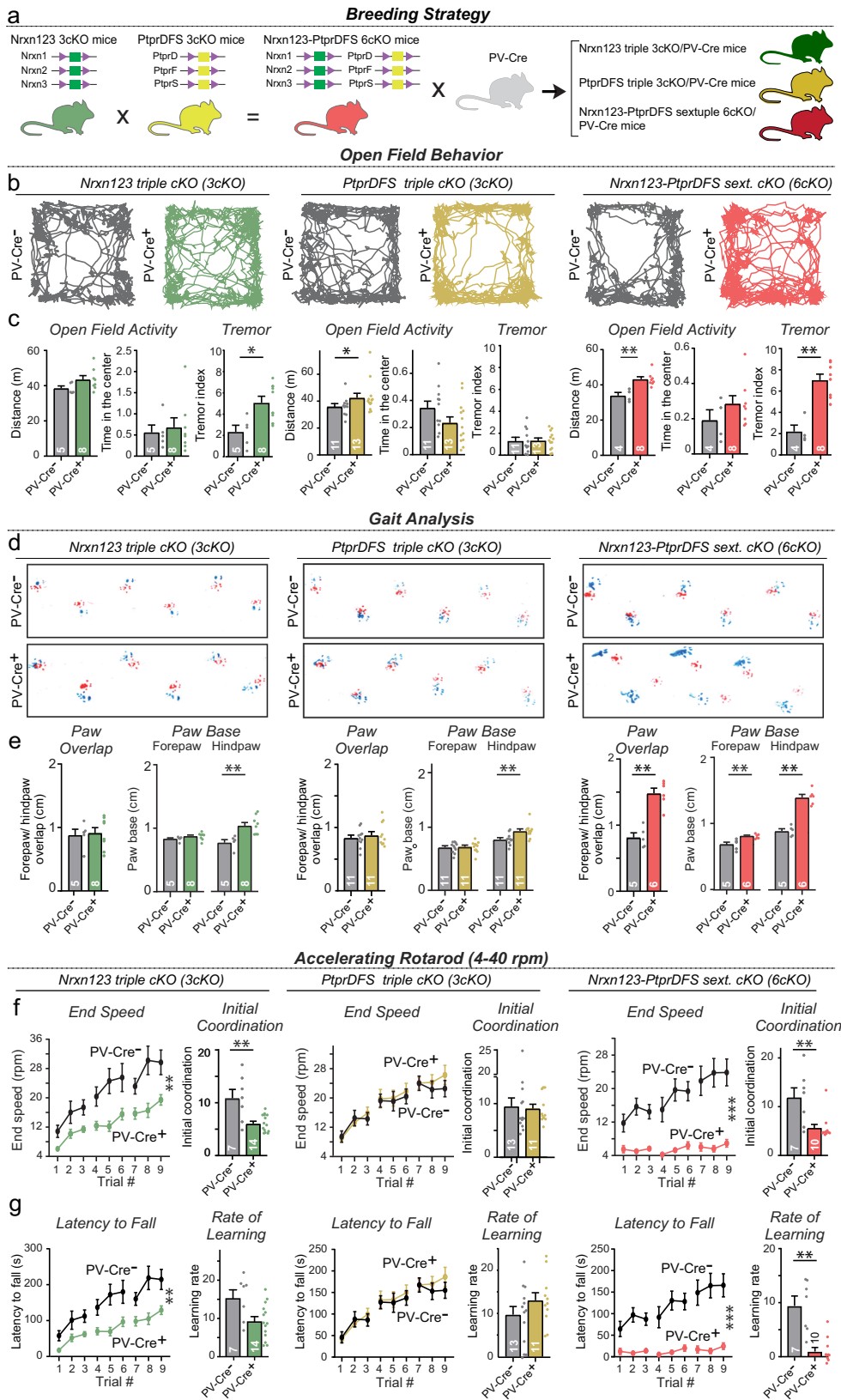

PC→DCN synapses, whereas separate deletion of neurexins or LAR-PTPRs has no major effect.

Immunocytochemical measurements of synapse density suffer from inherent limitations as they do not allow identification of the presynaptic origins of the synapses and do not provide information about synapse function. To overcome these limitations, we analyzed

PC→DCN synapses using retrograde pseudo-rabies virus tracing as an independent approach that enables the mapping of specific synaptic connections[46] (Fig. 5a–d). Neither *Nrxn123* 3cKO PV-Cre nor *PtprDFS* 3cKO PV-Cre mice exhibited a decrease in PC→DCN connections as monitored using pseudo-rabies virus tracing, whereas in *Nrxn123-PtprDFS* 6cKO mice, approximately 75% of synaptic connections were

**Fig. 2 | Triple conditional Nrxn123 or PtprDFS 3cKO mice crossed with PV-Cre driver mice exhibit only minor motor impairments, whereas sextuple conditional Nrxn123-PtprDFS 6cKO mice crossed with PV-Cre driver mice display a severe motor phenotype. a** Breeding strategy. Triple Nrxn123 and PtprDFS cKO mice were crossed to PV-Cre mice and to each other to generate Nrxn123 3cKO/PV-Cre, PtprDFS 3cKO/PV-Cre, and sextuple Nrxn123-PtprDFS cKO/PV-Cre mice with littermate controls lacking PV-Cre that were used in all experiments.
**b** Representative traces of locomotor activity of Nrxn123 triple cKO (left), PtprDFS triple cKO (middle), or Nrxn123, PtprDFS sextuple cKO (right) mice tested in the open field arena. **c** Quantification of the open field test and tremor behavior. Summary graphs show data for the distance travelled, and the time spend in the center of the arena. Graphs on the right illustrate the tremor behavior of mice, analyzed using 10 min force plate measurements[44, 68]. The tremor index was calculated by integrating the power in the 9-Hz to 12-Hz range. The averaged power in the 3-Hz to 6-Hz range was used as baseline. **d** Representative images from

footprint analyses comparing the gait of PV-Cre- and PV-Cre+ littermates from Nrxn123 triple cKO (left), PtprDFS triple cKO (middle), or Nrxn123, PtprDFS sextuple cKO (right) mice. **e** PV-Cre+ sextuple cKO mice exhibit less accurate and uniform foot placement compared to PV-Cre-, as evidenced by an increase in the distance between the placement of the right forepaw and hindpaw, and left forepaw and hindpaw (forepaw/hindpaw overlap), as well as changes in the hindpaw footprint size. **f, g** Performance of littermate PV-Cre- and PV-Cre+ mice on the accelerating rotarod (4–40 rpm). Latency to fall and end speeds are plotted across trials. Summary graphs show initial coordination and learning rate confirming severe motor impairments in mice lacking both Nrxn123 and PtprDFS. All data in summary plots and summary graphs are means ± SEM. Statistical significance was assessed by two-tailed Mann-Whitney test for summary graphs, or two-way ANOVA for graphs in **f**, **g**. All graphs show independent replicates (n = mice). Source data and statistical results are provided within the Source Data file.

---

lost (Fig. 5c, d). These results confirm that neurexins and LAR-PTPRs are indeed functionally redundant in establishing and maintaining PC→DCN synapses.

To further analyze the impairment induced by deletions of neurexin and LAR-PTPR, we examined acute cerebellar slices from *Nrxn123* 3cKO PV-Cre, *PtprDFS* 3cKO PV-Cre and *Nrxn123-PtprDFS* 6cKO mice and their littermates by electrophysiology (Fig. 6). Strikingly, again neither *Nrxn123* 3cKO PV-Cre nor *PtprDFS* 3cKO PV-Cre mice exhibited a major decrease in mIPSC frequency or amplitude monitored in DCN neurons (Fig. 6b, c, S7), a result that is consistent with previous studies at other synapses[29–33]. Neurons in the deep cerebellar nuclei of *Nrxn123-PtprDFS* 6cKO mice, however, displayed an approximately 70% decrease in mIPSC frequency without a major change in mIPSC amplitude (Fig. 6c, S7). Measurements of evoked IPSCs confirmed a large decrease in PC→DCN synaptic connectivity that manifested in a lowering of the IPSC amplitude (Fig. 6d–f). Consistent with a loss of synapses, no change in paired-pulse ratio was detected (Fig. 6g). Thus, the deletion of both neurexins and LAR-PTPRs, but not the deletion of either neurexins or LAR-PTPRs separately, produces a loss of at least half of PC→DCN synapses.

## Discussion

Our study addresses a fundamental conundrum: Given that neurexins and LAR-PTPRs together account for the majority of known transsynaptic adhesion complexes (Fig. 1a), why are these presynaptic 'hub' adhesion molecules not essential for synapse formation? Specifically, we and others described major changes in synaptic transmission upon genetic manipulations of various neurexins or LAR-PTPRs, but no consistent loss of synapses was observed[29–33,47]. Here, we addressed this question by generating sextuple conditional KO mice that target all major neurexin and LAR-PTPR isoforms (referred to as *Nrxn123-PtprDFS* 6cKO mice). We then analyzed the effect of the sextuple neurexin and LAR-PTPR deletions on PC→DCN synaptic connections as a model synapse, comparing sextuple KOs to triple KOs of only neurexins or LAR-PTPRs. Our data demonstrate that neurexins and LAR-PTPRs are functionally redundant at PC→DCN synapses despite the fact that they have no structural similarity or shared ligands except for Neuroligin-3[24]. The evidence for this conclusion consists of five sets of observations.

First, we find that most PC→DCN synapses (>85%) contain at least one nanocluster of a neurexin and of PtprS as analyzed separately with antibodies that label all neurexins or PtprS (Fig. 1). This result indicates that more than 70% of synapses of synapses contain both neurexins and LAR-PTPRs. The finding that neurexins and PtprS are present in one or multiple nanoclusters at a synapse agrees well with the emerging view that all synapses are non-uniform junctions that are organized in functional nanodomains[38–43,48–50].

Second, we show that the sextuple neurexin/LAR-PTPR deletions severely affect motor behaviors, whereas triple LAR-PTPR deletions

have no effect and triple neurexin deletions have a much lesser effect (Fig. 2). The lack of a significant change in the triple LAR-PTPR deletion except for a small gait impairment is surprising in view of the major functions ascribed to these presynaptic adhesion molecules. The triple neurexin deletion, conversely, on its own already elicited an essential tremor-like phenotype (Fig. 2c), consistent with previous findings that synaptic impairments in deep cerebellar nuclei cause tremor[45]. Moreover, the triple neurexin KO exhibited a modest phenotype on the accelerating rotarod (Fig. 2f, g). The sextuple neurexin/LAR-PTPR deletions greatly aggravated the triple neurexin KO phenotypes, and additionally elicited major open field and gait impairments (Fig. 2c, e–g). All of these impairments were not associated with any apparent change in the cerebellar architecture (Fig. 3).

Third, we demonstrate by confocal microscopy and dSTORM that the sextuple neurexin/LAR-PTPR deletions cause a robust loss of PC→DCN synapses (50–60% loss of synapses based on dSTORM analyses), whereas the triple neurexin and LAR-PTPR deletions do not (Fig. 4). The lack of a synapse loss in the neurexin and LAR-PTPR triple deletions confirms previous observations[29–33,51].

Fourth, we document by retrograde pseudo-rabies virus tracing that the sextuple neurexin/LAR-PTPR deletions severely impair PC→DCN synaptic connectivity (~75% loss of synaptic connections), whereas again the triple neurexin and LAR-PTPR deletions had no effect (Fig. 5). Importantly, the apparent loss of synaptic connectivity monitored via retrograde pseudotyped-rabies virus tracing is more severe than the decrease in synapse numbers determined morphologically (Fig. 4), suggesting that some of the synapses identified by immunocytochemistry may be non-functional.

Fifth and finally, we show by slice electrophysiology that the triple neurexin and the triple LAR-PTPR deletions have no significant effect in the mIPSC frequency in deep cerebellar nuclei neurons, whereas the sextuple neurexin/LAR-PTPR deletions greatly decreased the mIPSC frequency (an approximately 70% decrease; Fig. 6a–c). None of these genetic manipulations altered the mIPSC amplitude. Evoked IPSC recordings confirmed a large decrease in the strength of PC→DCN synapses without a change in paired-pulse ratio, consistent with a loss of synapses (Fig. 6d–g).

Overall, these findings establish that the sextuple neurexin/LAR-PTPR deletion induces phenotypes at PC→DCN synapses that greatly exceed those of the triple neurexin or triple LAR-PTPR deletions – in fact, the latter do not seem to cause any significant phenotypes at these synapses as monitored by our analysis. Our use of four independent methods of assessing phenotypes at PC→DCN synapses (motor behavior, synapse quantifications, retrograde pseudorabies virus tracing, and electrophysiology) ensures that the overall conclusion of functional redundancy among neurexins and LAR-PTPRs is correct. It is noteworthy that the relative magnitude of phenotypes differs between the approaches we used to monitor synapses, possibly because the sextuple neurexin/LAR-PTPR

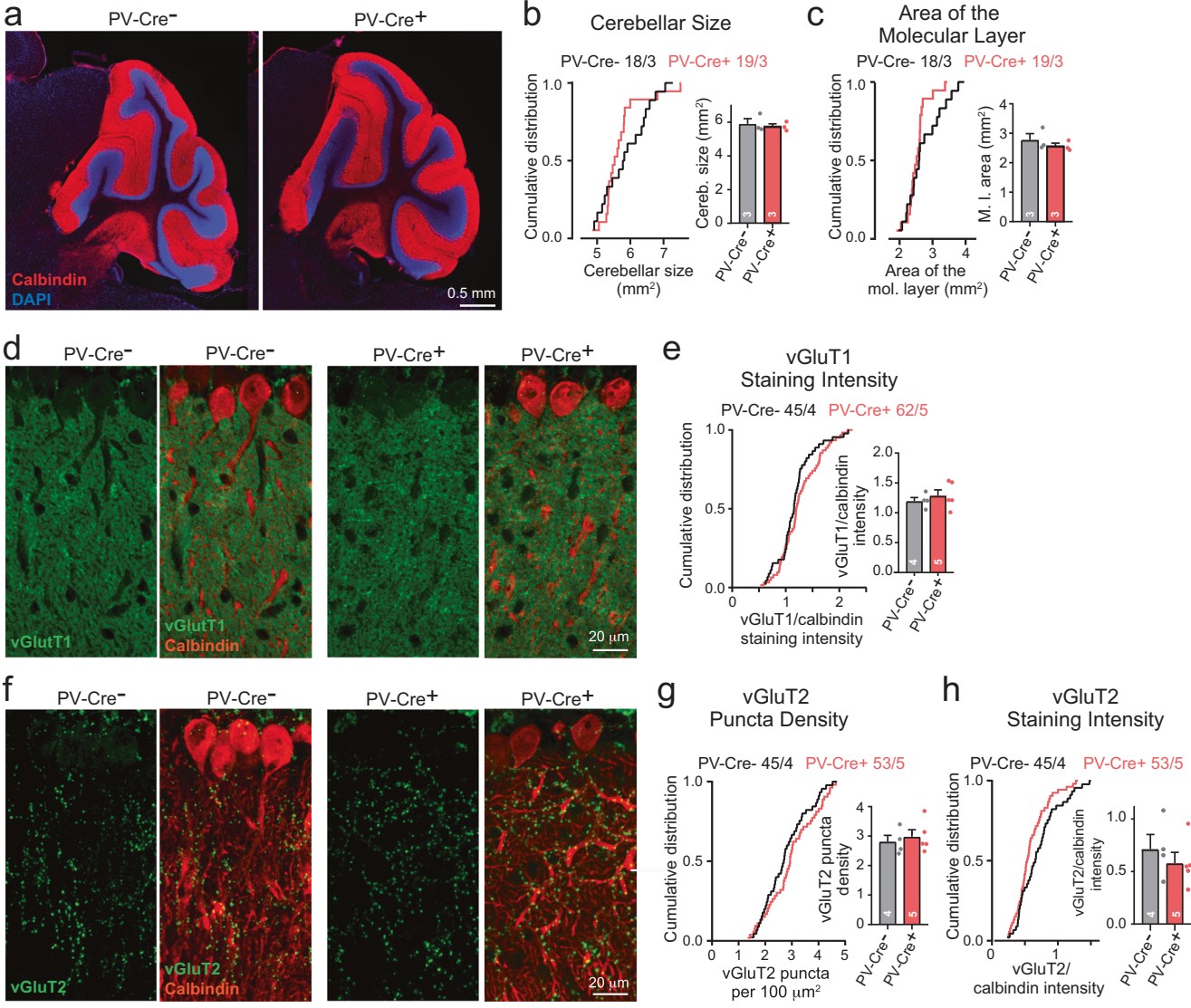

**Fig. 3 | Sextuple cKO mice targeting all major neurexin and LAR-PTPR isoforms do not exhibit major alterations in cerebellar cytoarchitecture.**
**a** Representative low-power images of the cerebellum, immunostained with antibodies to Calbindin (red) as a marker of Purkinje cells and with DAPI (blue). **b**, **c** Cumulative plots and summary graphs showing that deletion of Nrxn123 and PtprDFS in PV neurons does not affect the size of the cerebellum (**b**) or the thickness of the cerebellar cortex (**c**). **d** Representative confocal images of Purkinje cells immunostained with antibodies to Calbindin (red) and vGluT1 (green) as a marker of parallel-fiber synapses. **e** Cumulative plots and summary graphs for independent replicates of images in **d** showing that deletion of Nrxns and LAR-PtprDFS in PV neurons does not affect the vGluT1 staining signal as a proxy of parallel-fiber excitatory synapses in the cerebellum. **f** Representative confocal

images of Purkinje cells immunostained with antibodies to Calbindin (red) and vGluT2 (green) as a marker for climbing-fiber synapses. **g**, **h** Cumulative plots and summary graphs of the vGluT2-positive puncta density (**g**) or the vGluT2 staining intensity (**h**) as a measure of climbing fiber synapse densities. All data in cumulative plots represent data from individual images ($n$ = ROIs/mice) whereas all data in summary graphs represent true replicates ($n$ = mice) and are shown as means ± SEM. Statistical significance was assessed by Kolmogorov-Smirnov test for cumulative plots and by two-tailed Mann-Whitney test for summary graphs. For **b**, **c**, $n$ = ROIs/mice: Nrxn123, PtprDFS 6cKO mice ($n$ = 18/3 for PV-Cre-, $n$ = 19/3 for PV-Cre + ); for **e**–**h**, $n$ = ROIs/mice: Nrxn123, PtprDFS 6cKO mice ($n$ = PV-Cre- $n$ = 45/4, PV-Cre+ $n$ = 53/5). Source data and statistical results are provided within the Source Data file.

deletion decreases the number of synapses less than it impairs the function of synapses.

Our observations also raise new questions. Most important among these questions is probably that of the mechanism involved, given that neurexins and LAR-PTPRs do not interact with common postsynaptic ligands. Their only known shared postsynaptic interaction partner, Neuroligin-3[24], is not essential for synapse formation and deletion of Neuroligin-3 by itself has only discrete effects on a subset of synapses[52–59], suggesting that the interaction of both neurexins and LAR-PTPRs with Neuroligin-3[24] cannot account for their functional redundancy. The most parsimonious hypothesis that accounts for the redundancy of neurexins and LAR-PTPRs is that synapses are

assembled via multiple parallel trans-synaptic interactions that are not all required for making a synapse as such at any given time but uniquely perform other core functions. Each of these parallel trans-synaptic interactions may be necessary for enabling particular properties of synapses, with only their combinatorial action being required for synapse assembly. For example, LAR-PTPRs and neurexins separately regulate specific features of NMDA-receptors at synapses[33,48,52,53], and neurexins are essential for organizing presynaptic calcium channels[27,30,31], even though separately neither one is required for synaptic connectivity overall. An alternative hypothesis that is equally plausible would be that the loss of synaptic connectivity we observed upon deleting both neurexins and LAR-PTPRs is due to a mass action

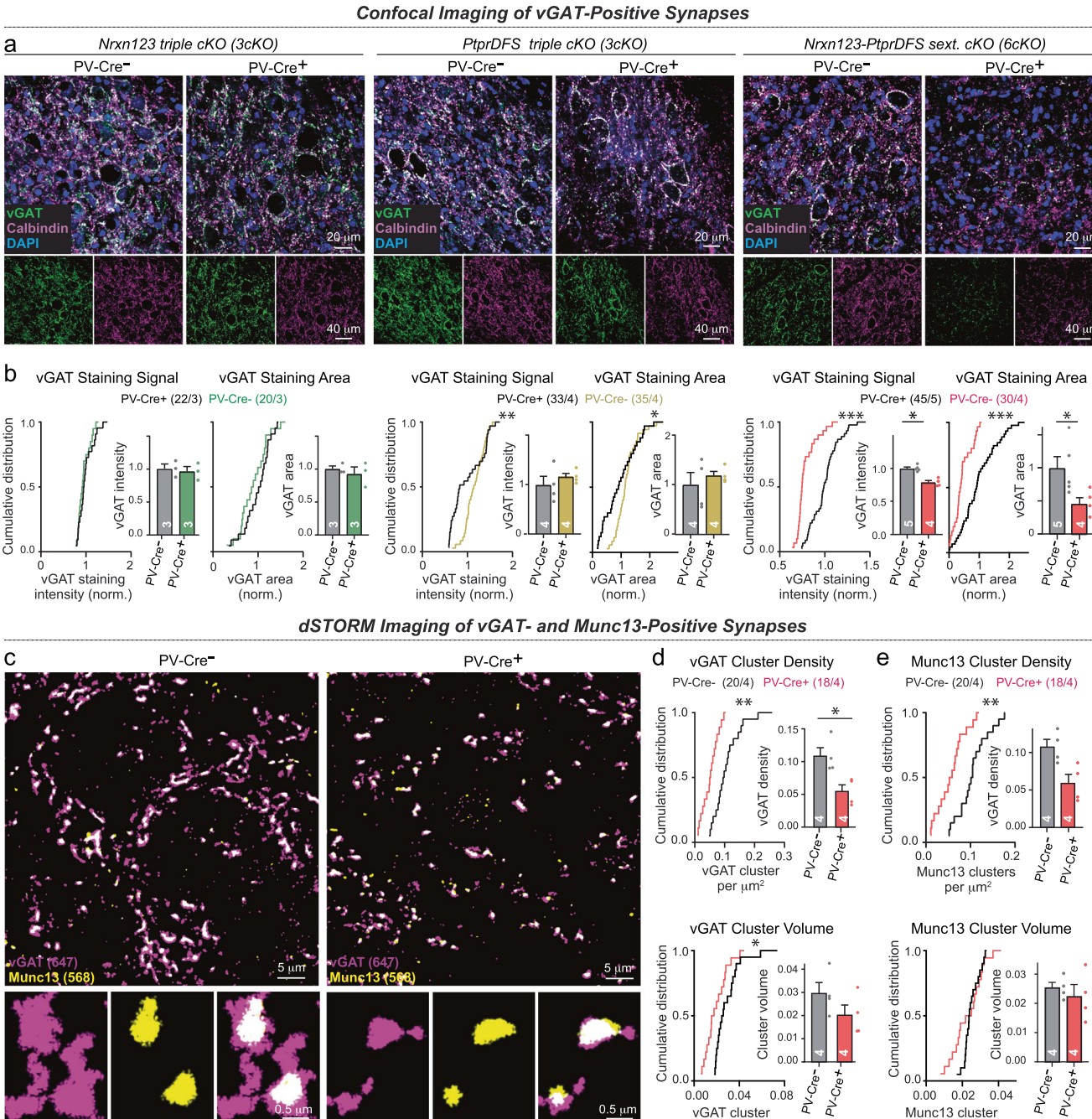

**Fig. 4 | Triple conditional Nrxn123 or PtprDFS 3cKO mice crossed with PV-Cre driver mice exhibit no loss of inhibitory synapses in deep cerebellar nuclei, whereas sextuple conditional Nrxn123-PtprDFS 6cKO mice crossed with PV-Cre driver mice feature a large decrease in synapse density. a** Representative confocal images of the deep cerebellar nuclei (DCN) immunostained for vGAT (green), Calbindin (magenta), and DAPI (blue) to visualize inhibitory Purkinje-cell synapses formed on deep cerebellar nuclei neurons. **b** Cumulative plots of individual data points and summary graphs of the vGAT-staining signal and area reveal that triple neurexin and LAR-PTPR deletions cause no apparent change in synaptic vGAT signals whereas simultaneous sextuple deletions of neurexins and LAR-PTPRs induce a significant reduction in vGAT-staining intensity and area, suggesting a structural loss of inhibitory synapses. **c** Representative dSTORM images of sections of the deep cerebellar nuclei stained for vGAT (magenta, 647 fluorophore) and Munc13 (yellow, 568 fluorophore), illustrating coincident synaptic clusters stained for these two synapse markers. **d**, **e** Quantification of the density (top) and volume (bottom) of synaptic clusters visualized by staining for vGAT (**d**) or Munc13 (**e**) in the same sections of deep cerebellar nuclei. Note the large reduction in cluster density without a major change in cluster volume. All data in summary graphs are means ± SEM. Statistical significance was assessed by Kolmogorov-Smirnov test for cumulative plots and by two-tailed Mann-Whitney test for summary graphs. For **b** *n* = ROIs/mice: Nrxn123 3cKO (*n* = 22/3 for PV-Cre-, *n* = 20/3 for PV-Cre + ); PtprDFS 3cKO mice (*n* = 33/4 for PV-Cre-, *n* = 35/4 for PV-Cre + ); Nrxn123, PtprDFS 6cKO mice (*n* = PV-Cre- *n* = 45/5, PV-Cre+ *n* = 30/4). For **d**–**e**, *n* = ROIs/mice: Nrxn123, PtprDFS 6cKO mice (*n* = PV-Cre- *n* = 20/4, PV-Cre+ *n* = 18/4). Source data and statistical results are provided within the Source Data file.

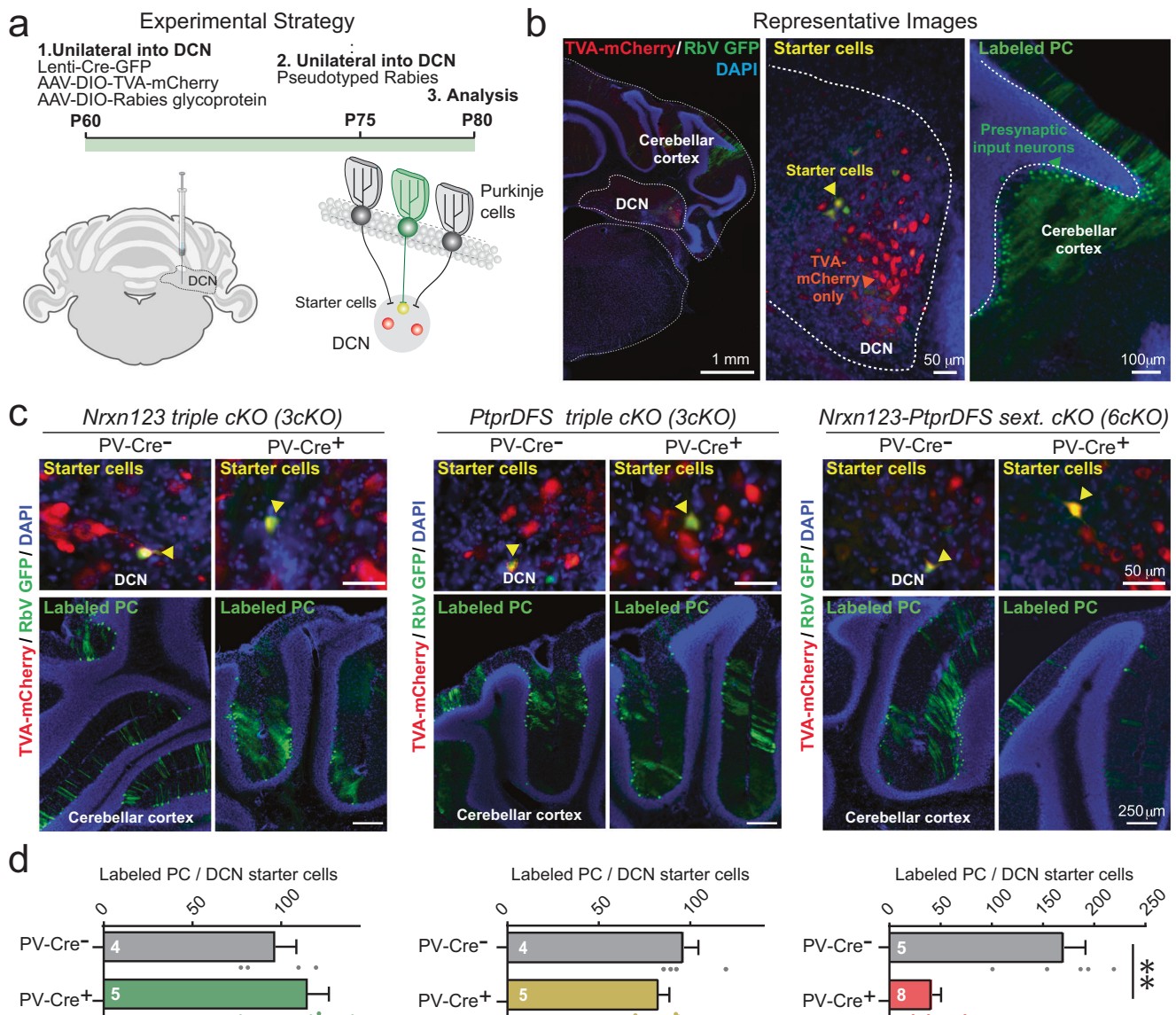

**Fig. 5 | Retrograde pseudo-rabies virus tracing reveals major loss of synaptic connections in sextuple conditional *Nrxn123-PtprDFS* 6cKO mice crossed with PV-Cre driver mice, whereas the corresponding triple conditional *Nrxn123* or *PtprDFS* 3cKO mice crossed with PV-Cre driver mice exhibit only minor changes. a** Experimental strategy for **b**–**d**. Purkinje cells to deep cerebellar nuclei connectivity was probed by retrograde monosynaptic pseudo-rabies virus tracing. **b** Representative images of a monosynaptic pseudo-rabies virus tracing experiment, showing an overview of the cerebellum (left), a magnification of the deep cerebellar nuclei containing TVA-mCherry positive cells (red) and starter cells (infected with AAVs encoding for pseudo-rabies-complementing proteins and pseudo-typed rabies virus, yellow, middle panel), as well as a magnification of the cerebellum showing retro-labeled Purkinje cells (green, right). **c** Representative images of starter cells in the deep cerebellar nuclei (top panel) and retro-labeled Purkinje cells (green, bottom panel) for PV-Cre- and PV-Cre+ *Nrxn123* 3cKo mice, *PtprDFS* 3cKO mice and *Nrxn123-PtprDFS* 6cKO mice. **d** Quantification of synaptic connections of Purkinje cells on deep cerebellar nuclei as measured by retrograde pseudo-rabies virus tracing reveals a major phenotype only in *Nrxn123-PtprDFS* 6cKO but not in *Nrxn123* 3cKo or *PtprDFS* 3cKO mice. Data are means ± SEM. Statistical significance was assessed by two-tailed Mann-Whitney test (Nrxn123 3cKo mice PV-Cre- *n* = 4 mice & PV-Cre+ *n* = 5 mice; PtprDFS 3cKO mice PV-Cre- *n* = 4 mice & PV-Cre+ *n* = 5 mice; Nrxn123-PtprDFS 6cKO mice PV-Cre- *n* = 5 mice, PV-Cre+ *n* = 8 mice). Source data and statistical results are provided within the Source Data file.

effect. According to this hypothesis, the ablation of many different trans-synaptic interactions in the sextuple neurexin and LAR-PTPR deletion may simply overwhelm the ability of synapses to cope with the loss of multiple separate regulatory processes, suggesting that the apparent redundancy between neurexins and LAR-PTPRs is not a functional but a structural redundancy. One could argue that these two hypotheses are related since both posit that the redundancy is due to the loss of a large number of trans-synaptic interactions.

Another question raised by our findings is how the redundancy of neurexins and LAR-PTPRs in synapse assembly relates to the role of *Nrxn2* in restricting synapses? We recently showed that at least in hippocampal Schaffer-collateral synapses, deletion of *Nrxn2* causes a surprising increase in synapse numbers and in release probability[60]. This finding indicated that at these synapses, *Nrxn2* functions to limit instead of promoting the number of synapses. At least two explanations could account for this observation in the context of the sextuple neurexin/LAR-PTPR deletion. Plausibly in the triple neurexin deletions opposing actions of different neurexins on synapse numbers could cancel each other out, or alternatively neurexins have both pro- and anti-synaptogenic functions that are differentially regulated by alternative splicing, which may differ between synapses. Future experiments will have to address these and other hypotheses.

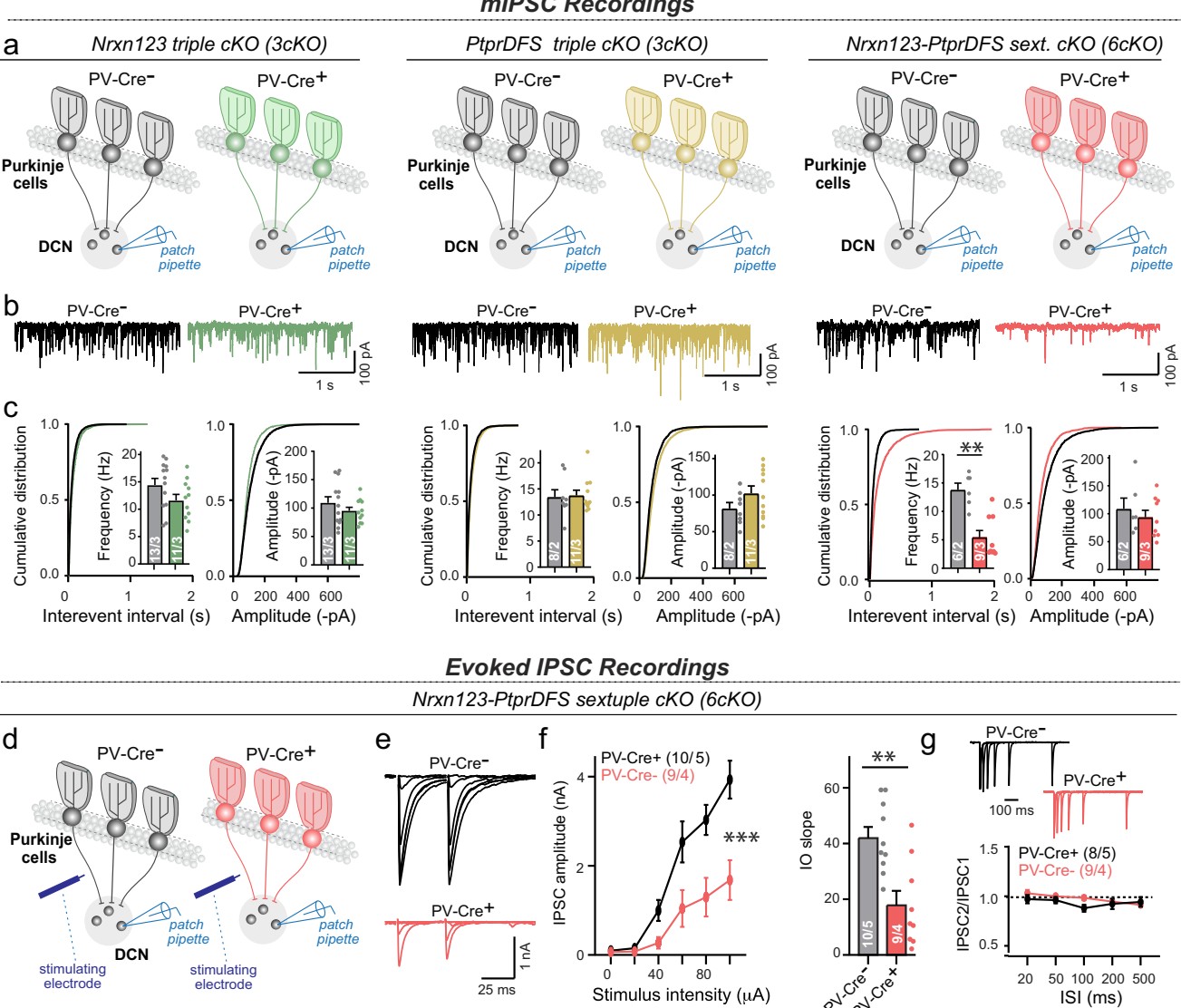

**Fig. 6 | Triple conditional Nrxn123 or PtprDFS 3cKO mice crossed with PV-Cre driver mice exhibit no major changes in synaptic function, whereas sextuple conditional Nrxn123-PtprDFS 6cKO mice crossed with PV-Cre driver mice display a severe decrease in synaptic strength.** a–c Analysis of mIPSCs in triple *Nrxn123* (left) and triple *PtprDFS* 3cKO mice (middle) fails to uncover a major change, whereas the mIPSC frequency is severely decreased in sextuple *Nrxn123-PtprDFS* 6cKO mice (right) (**a**, schematic of · whole-cell patch-clamp recordings performed in acute deep cerebellar nuclei slice from P21-P26 mice; **b** mIPSC sample traces; **c** cumulative probability plots and summary graphs of the mIPSC interevent intervals and frequency, respectively, and cumulative probability plots and summary graphs of the mIPSC amplitudes). **d–g** Analysis of evoked IPSCs reveals that the *Nrxn123-PtprDFS* 6cKO causes a major decrease in synaptic strength of Purkinje cell synapses on deep cerebellar nuclei (**d** schematic of the recording paradigm in which evoked IPSCs were induced in acute slices by electrical stimulation of Purkinje cell axons and

monitored in DCN neurons; **e** sample traces; **f** summary plot of evoked IPSCs as a function of the stimulus intensity and summary graph of the slope of input-output curves of evoked IPSCs recorded in deep cerebellar nuclei neurons; **g** sample traces of paired-pulse measurements of evoked IPSCs (top) and summary graph of the paired-pulse ratio of evoked IPSCs as a function of the inter-stimulus intervals (bottom) recorded in deep cerebellar nuclei neurons. All data in summary graphs and plots are means ± SEM. Statistical significance was assessed by Kolmogorov-Smirnov test for cumulative plots, two-tailed Mann-Whitney test for summary graphs, and two-way ANOVA for summary plots in **f** and **g**. For **a**–**c** n = cells/mice: Nrxn123 3cKO mice (PV-Cre- n = 13/3, PV-Cre+ n = 11/3); PtprDFS 3cKO mice (PV-Cre- n = 8/2, PV-Cre+ n = 11/3); Nrxn123, PtprDFS 6cKO mice (PV-Cre- n = 6/2, PV-Cre+ n = 9/3). For **f**, n = cells/mice: Nrxn123, PtprDFS 6cKO mice (PV-Cre- n = 10/5, PV-Cre+ n = 9/4); for **g**, n = cells/mice: Nrxn123, PtprDFS 6cKO mice (PV-Cre- n = 8/5, PV-Cre+ n = 9/4). Source data and statistical results are provided within the Source Data file.

In addition to these questions, several limitations of our study should be noted. First, although we deleted the vast majority of neurexins and LAR-PTPRs, one neurexin isoform remains expressed: Nrxn1γ[21]. We cannot exclude a large effect of this short variant of *Nrxn1* especially since it has been implicated in synapse formation in C. elegans[61], but three considerations argue against a major role for Nrxn1γ in mouse neurons. First, Nrxn1γ expression levels in neurons and glia are low[21]. Second, Nrxn1γ lacks all extracellular domains of neurexins that are known to interact with trans-synaptic ligands,

suggesting that any role it might have would have to operate via presynaptic cis-interactions. Third, a *Nrxn1* deletion that also targets Nrxn1γ exhibits no major synapse loss phenotype in mice[38].

Another limitation of this study is that we analyzed only a single type of synapse in a single brain region, PC→DCN synapses. It is possible that at other synapses, the role of neurexins and LAR-PTPRs differs. Our study only establishes the redundancy of neurexins and LAR-PTPRs at PC→DCN synapses, and more studies are required to explore the generality of our conclusion.

In summary, we here demonstrated that combinatorial expression of neurexins and LAR-PTPRs collaborate in organizing functional PC→DCN synapses. Although the effect size of the phenotypes we observed upon deletion of Neurexins and LAR-PTPRPs is very large, our analysis indicated that some synapses are still forming, suggesting that synapses are built by the collaborative efforts of an even larger number of SAMs than those encoded by neurexin and LAR-PTPR genes. Elucidating the rules that govern synapse assembly, both their initial establishment and the shaping of their functional properties, is a fascinating goal that will need to be reached for any understanding of how neural circuits are constructed in the brain.

## Methods

This research complies with ethical regulations. All in vivo procedures conformed to National Institutes of Health Guidelines for the Care and Use of Laboratory Mice and were approved by the Stanford Animal Use Committees [Administrative Panel for Laboratory Animal Care (APLAC/) Institutional Animal Care and Use Committee (IACUC)], under the animal protocols 18846 and 20787.

### Animals

LAR-PTPR triple cKO mice and Nrxn123 triple cKO mice were generated as previously described[30,33]. Briefly, Nrxn123 triple cKO mice were generated by flanking exon 18 with FloxP site[30]. PtprDFS 3cKO mice were obtained by crossing Ptprd cKO mice (Ptprdtm2a(KOMP)Wtsi, colony prefix MEXY, ESC clone ID: EPD0581_9_D04, RRID: IMSR_EM:11805, then crossed to Flp mice (Jackson Laboratory, JAX:005703, RRID: IMSR_JAX:005703), Ptprs cKO mice (Ptprs_tm1c_D11, ES cell clone ID: DEPD00535_1_D11, RRID: IMSR_KOMP:CSD76529-1c-Mbp), and PTPRF cKO mice (generated by flanking exon 4 with loxP sites)[33].

Nrxn123-PtprDFS sextuple 6cKO mice were generated by crossing Nrxn123 3cKO and PtprDFS 3cKO mice over multiple generations. PV-Cre (Jackson Lab JAX # 017320) mice were included in these crosses to generate the Nrxn123-PtprDFS 6KO/PV-Cre mice. C57BL/6 J (Jackson Laboratory, JAX #000664), PV-Cre L7-cre (Jackson Laboratory, JAX #004146), and Ribotag (Jackson Laboratory, JAX #029977) mice were purchased from the Jackson laboratory. Mice were group-housed on a 12 h light-dark cycle with access to food and water ad libitum. Male and female mice were used for all experiments.

### Plasmids

The following plasmids were used: lentiviral vectors expressing EGFP tagged Cre recombinase under the control of the synapsin promoter (FSW-NL-EGFP-CRE); lentivirus helper plasmids (VSVG expression vector, pRRE and pRSV-REV), pAAV-CAG-EGFP-CRE and pAAV-CAG-EGFP-ΔCRE, AAV-DJ helper plasmids (pHelper and pRC-DJ).

### Source of RNAseq data

Expression of presynaptic proteins in different brain regions was downloaded directly from the Dropviz website (http://dropviz.org/), which reports Drop-seq data of 690,000 individual cells from nine different regions of the adult mouse brain. For expression in the DCN, data were obtained from ref. [62]. (https://www.ncbi.nlm.nih.gov/geo/query/acc.cgi?acc = GSE160471).

### Virus production

For the production of lentiviruses, the lentiviral expression shuttle vector and three helper plasmids (pRSV-REV, pMDLg/pRRE and vesicular stomatitis virus G protein, VSVG) were co-transfected into HEK293T cells (ATCC, CRL-11268), at 5 μg of each plasmid per 25 cm² culture area, respectively. Transfections were performed using the calcium-phosphate method. Media with viruses was collected at 48 hr after transfection, filtered (0.45 μm pore size), concentrated by ultracentrifugation, and used for in vivo experiments. Complementing AAVs containing CAG-FLEX-TCB-mCherry in capsid 2/5and CAG-FLEX-

RG in capsid 2/8, as well as bV-CVS-N2c-deltaG-GFP (EnvA) were generated at the Janelia Farm Viral core facility.

### Purification of Ribosome-bound mRNA

Ribosome-bound mRNA was purified as described previously[62] with minor modifications. L7-cre mice crossed to ribotag mice were euthanized using isoflurane and decapitated. 4 mice used in total (2 males, 2 females). The cerebellum was quickly dissected and snap-frozen in liquid nitrogen and transferred to −80 °C storage until processing. Frozen brains were partially thawed in fresh homogenization buffer at 10% weight/volume and Dounce homogenized. Homogenate was clarified by centrifugation and 10% of the supernatant was saved as input. The remaining lysate was incubated with pre-washed anti-HA magnetic beads (Thermo Fisher, 88837) overnight at 4 °C. The beads were washed 3 times with a high-salt buffer followed by elution with RLT lysis buffer with β-ME. Both input and IP samples were subjected to RNA extraction using the QIAGEN RNeasy Micro kit. RNA concentration was determined using a NanoDrop 1000 Spectrophotometer and stored at −80 °C until downstream analysis.

### Quantitative RT-PCR

Qantitative RT-PCR was run in QuantStudio 3 (Applied biosystems, Thermo Fisher Scientific, USA) using TaqMan Fast Virus 1-Step Master Mix (PN4453800, Applied biosystems, Thermo Fisher Scientific, USA). For measuring the purity of cell type-specific ribosome-bound mRNA following immunoprecipitation, the following predesigned assays were used (gene, assay ID): Actb (Mm.PT.51.14022423), Calb1 (Mm.PT.58.29798692), Pcp2 (Mm.PT.58.30961208).

To quantify the mRNA levels of Nrxn and LAR-Ptpr isoforms, the same mRNAs were probed using the following assays (gene, primer 1, primer 2, probe): Nrxn1α (5′-TTCAAGTCCACAGATGCCAG-3′, 5′-CAAC ACAAATCACTGCGGG-3′, 5′-TGCCAAAAC/ZEN/TGGTCCATGCCAAA G-3′); Nrxn1β (5′-CCTGTCTGCTCGTGTACTG-3′, 5′-TTGCAATCTACAGG TCACCAG-3′, 5′-FAM/AGATATATG/ZEN/TTGTCCCAGCGTGTCCG-3′); Nrxn1γ (5′-GCCAGACAGACATGGATATGAG-3′, 5′-GTCAATGTCCTCA TCGTCACT-3′, 5′-ACAGATGAC/ZEN/ATCCTTGTGGCCTCG-3′); Nrxn2α (5′-GTCAGCAACAACTTCATGGG-3′, 5′-AGCCACATCCTCACAACG-3′, 5′-FAM/CTTCATCTT/ZEN/CGGGTCCCCTTCCT-3′); Nrxn2β (5′-CCACCAC TTCCACAGCAAG-3′, 5′-CTGGTGTGTGCTGAAGCCTA-3′, 5′-GGACCAC AT/ZEN/ACAT CTTCGGG-3′); Nrxn3α (5′-GGGAGAACCTGCGAAAGA G-3′, 5′-ATGAAGCGGAAGGACACATC-3′, 5′-CTGCCGTCA/ZEN/TAGCTC AGGATAGATGC-3′); Nrxn3β (5′-CACCACTCTGTGCCTATTTC-3′, 5′-GG CCAGGTATAGAGGATGA-3′, 5′-TCTATCGCT/ZEN/CCCCTGTTTCC-3′); PTPRS (Mm.PT.58.30580505); PTPRF (Mm.PT.58.14060589); PTPRD (Mm.PT.58.45964964).

### Immunochemistry

Mice were anesthetized and sequentially perfused with phosphate buffered saline (PBS) and ice cold 4% paraformaldehyde (PFA). Brains were dissected and post-fixed in 4% PFA overnight, then cryoprotected in 30% sucrose in PBS for 24 h. 40 μm thick coronal sections of the cerebellum containing the DCN were collected using a Leica CM3050-S cryostat (Leica, Germany). Free floating brain sections were incubated with blocking buffer (5% goat serum, 0.3% Triton X-100 in PBS) for 1 h at room temperature, then treated with primary antibodies diluted in blocking buffer overnight at 4 °C (anti-vGluT1, AB5905, Millipore Sigma, 1:1,000; anti-vGluT2, AB2251-I, Millipore Sigma, 1:1,000; anti-vGAT, 131004, Sysy,1:1,000; anti-calbindin, C9848, Millipore Sigma). Sections were washed three times with PBS (15 min each), then treated with secondary antibodies for 1 h at room temperature. After washing with PBS 4 times (15 min each), sections were mounted onto Superfrost Plus slides with mounting media containing DAPI. Confocal images were acquired with a Nikon confocal microscope (A1Rsi, Nikon, Japan) with 60x oil objective, at 1024 ×1024 pixels, with z-stack distance of 0.3 μm. All acquisition parameters were kept constant within the same day

between groups. Images were analysed with Nikon analysis software. Multiple parameters were analysed to assess potential phenotypes. These include staining intensity, a common measurement which correlate with protein abundance as well as puncta density (when individual synaptic puncta could be resolved) or staining area (when individual puncta were not resolved) to measure the number of synapses.

### Direct Stochastic Optical Reconstruction Microscopy (dSTORM)

dSTORM images were acquired using a Vutara SR 352 (Bruker Nanosurfaces, Inc., Madison, WI) commercial microscope based on single molecule localization biplane technology[63,64]. Mice were anesthetized and sequentially perfused with phosphate buffered saline (PBS) and ice cold 4% paraformaldehyde (PFA). Brains were dissected, trimmed to only include the cerebellum and brainstem area, and post-fixed in 4% PFA for 30 min, then cryoprotected in 30% sucrose in PBS for 48 h. 15–20 µm thick coronal sections of the cerebellum containing the DCN were collected using a Leica CM3050-S cryostat (Leica, Germany). Free floating brain sections were incubated with blocking buffer and labelled with a PAN-NRXN antibody (ABN161-I, Millipore Sigma, 1:300), anti-PTPRS (PAC9986, homemade, 1:300), anti-vGAT (131004, Sysy, 1:1000), MUNC-13 (126103, Sysy, 1:1000) primary antibodies and secondary antibodies conjugated to Alexa647 (1:1000/5000, Thermo-Fisher) or CF568 (1:1000/5000, Biotium). The slices were mounted on a coverslip coated with poly-L-Lysine. After drying, sections were briefly re-hydrated and post-fixed using 2% PFA for 10–15 min followed by three washes with DBS. Sections were stored at 4 C in DPBS shielded from light, until imaging. Coverslips were placed in dSTORM buffer containing (in mM) 50 Tris-HCl at pH 8.0, 10 NaCl, 20 MEA, 1% β-mercaptoethanol, 10% glucose, 150 AU glucose oxidase type VII (Sigma Cat#: G2133), and 1500 AU catalase (Sigma Cat#: C40). Labeled proteins were imaged with 647 and 561 nm excitation power of 40 kW/cm2. Images were recorded using a 60×/1.2 NA Olympus water immersion objective and Hamamatsu Flash4 sCMOS camera with the gain set at 50 and frame rate at 50 Hz. Data was analysed by the Vutara SRX software (version 6.04).

Munc13 and vGAT were imaged for 7000 frames with the first 1000 frames excluded from analysis. All other signals were imaged for 10000 frames with the first 3000 frames excluded from analysis. Identified molecules were localized in three dimensions by fitting the raw data in a 12 × 12-pixel region of interest centered around each particle in each plane with a 3D model function that was obtained from recorded datasets of fluorescent beads. Fit results were filtered by a density based denoising algorithm to remove isolated particles and rendered as 50 nm points. The remaining localizations were classified into clusters by density-based spatial clustering of applications with noise (DBSCAN), a minimum of 20 localizations were connected around a 100 nm search radius. The experimentally achieved image resolution of 40 nm laterally (x, y) and 70 nm axially (z) was determined by Fourier ring correlation.

### Monosynaptic retrograde rabies tracing

Mice were anesthetized with tribromoethanol (250 mg/kg, T48402, Sigma, USA), head-fixed with a stereotaxic device (KOPF model 1900). Sustained-release Buprenorphine was injected subcutaneously before the surgery as anti-analgesic treatment. Viruses were loaded via a glass pipette connected with a 10 µl Hamilton syringe (Hamilton, 80308, US) on a syringe injection pump (WPI, SP101I, US) and injected at a speed of 0.30 µl/min. A mixture of lentiviruses expressing synapsin-Cre-GFP and complementing AAVs containing CAG-FLEX-TCB-mCherry and CAG-FLEX-RG was injected in P60 mice. Mouse heads were shaved, the shaved area was cleaned with Betadine, lubricant was placed on the eyes (Puralube Vet Ointment). Coordinates used for unilateral DCN injections were AP − 6.2 mm, ML + 1.6 mm, DV − 3 mm. Mice were monitored in a warmed recovery cage until full recovery. RbV-CVS-N2c-deltaG-GFP (EnvA) was injected 2 weeks after AAV injections at an infectious titer of $1 \times 108$ IU/mL and 0.5 µL volume as described above. Mice were subsequently perfused and analyzed 5 days later. Brains were post-fixed overnight in 4% PFA/PBS and sliced on a vibratome in 50 µm sections. Sections were labelled with DAPI, washed with PBS, mounted, and imaged using the Olympus VS200 Slide Scanner at 20X.

### Preparation of acute brain slices for electrophysiology

Acute coronal brain slices containing the DCN were prepared from P21–26 mice. Isofluorane-anesthetized mice were decapitated, their brain removed and trimmed, and placed in a cold oxygenated (95% $O_2$, 5% $CO_2$) cutting solution that contained (in mM): 228 sucrose, 26 $NaHCO_3$, 11 glucose, 2.5 KCl, 1 $NaH_2PO_4$, 7 $MgCl_2$, 0.5 $CaCl_2$. 180 µm-thick slices were cut with a Leica vibratome (VT1200S) and recovered for 30 min at 32 °C in oxygenated ACSF solution containing (in mM): 119 NaCl, 2.5 KCl, 1.3 $MgCl_2$, 2.5 $CaCl_2$, 11 glucose, 1 $NaH_2PO_4$, and 26 $NaHCO_3$. Brain slices were then moved to a holding chamber filled with oxygenated ACSF at room temperature for 30 min.

### Electrophysiological recordings

DCN slices were moved to the recording chamber mounted onto an Axioskop FS-2 upright microscope (Zeiss). The microscope was equipped with DIC and fluorescence filters, and a LED source connected to the back port of the microscope via an optic fiber. Whole-cell voltage clamp recordings were performed on DCN neurons. Brain slices were maintained at -32 °C via a dual-T344 temperature controller (Warner Instruments). Brain slices were continuously perfused with normal oxygenated ACSF (at about 1 ml/min perfusion rate). Electrical signals were recorded at 25 kHz with a two channel Axoclamp 700B amplifier (Axon Instruments), digitalized with a Digidata 1440 digitizer (Molecular devices) that was in turn controlled by Clampex 10.7 (Molecular Devices). Synaptic currents were recorded using a pipette solution that contained (in mM): 135 CsCl2, 10 HEPES, 1 mM EGTA, 4 ATP-Na and 0.4 GTP-Na (300 mOsm l − 1, pH 7.3 adjusted with CsOH), and an external solution (standard ACSF) that contained (in mM): 119 NaCl, 2.5 KCl, 1.3 $MgCl_2$, 2.5 $CaCl_2$, 11 glucose, 1 $NaH_2PO_4$, and 26 $NaHCO_3$. The following pharmacological agents were used in the extracellular solution: CNQX (10µM, AMPAR blocker, Tocris Bioscience), AP5 (50µM, NMDAR blocker, Tocris Bioscience), TTX (1µM, voltage gated sodium channel blocker, American Radiolabeled chemicals). Recording pipettes were pulled from thin-walled borosilicate glass pipettes to resistances of 3–5 MΩ. Miniature IPSCs were recorded at holding potentials of −70mV in the presence of TTX (1µM, voltage gated sodium channel blocker, American Radiolabeled chemicals) in the bath solution.

Evoked synaptic currents were elicited with a bipolar stimulating electrode (A-M Systems, Carlsborg, WA), controlled by a Model 2100 Isolated Pulse Stimulator (A-M Systems, Inc.), and synchronized with the Clampfit 10 data acquisition software (Molecular Devices). Evoked IPSCs were recorded at holding potentials of −70mV, by electrical stimulation of Purkinje cells axonal boundaries. Paired-pulse ratios were monitored with interstimulus intervals of 20–500 ms. For all recordings and analyses, the experimenter was blind to genotype. Quality of the recordings was assessed during the experiments. Cells with unstable leak current, Ra, and Rin were not recorded or excluded from the analysis. All stable cells that had a Ra lower than 21 mOhm were recorded and included in the analysis.

### Force plate

5 min trials on a 28 × 28 cm force-plate actometer device were used to analyze the locomotor activity and the tremor level of the mice. For locomotor activity, changes in the center of force (movement) of the mouse was monitored and analyzed using an in-house program[65]. Total distance travelled and time spent in the center were plotted for every genotype.

Analysis of animal tremor was performed as previously described (45). Briefly, the high sampling rate (100 Hz) of weight measurement was used to develop a custom MATLAB script for data analysis. The raw data were divided into 3-second segments and fast Fourier transformation was performed for each 3-second segment. The power spectra were then averaged. A "tremor index" was calculated by integrating the power value in the 9-Hz to 12-Hz window. Power value in the 3-Hz to 6-Hz window was used as the baseline.

### Accelerating rotarod

Rotarod performance[56,66] was tested using a five-station rotarod treadmill (ENV-575M, Med Associates). The rod accelerated from 4 to 40 rpm at a constant rate of acceleration over 300 s. Testing consisted of three trials per day, separated by at least 30 min each, over the course of 3 days (9 total trials). Each trial was terminated when a mouse fell off, made one complete backward revolution while hanging on, or after 300 s (maximum speed, no further acceleration).

### Footprint analysis

Various gait parameters were quantified using footprint analysis[67]. A piece of white paper was placed along the floor of an open-top runway measuring 50 cm long, 10 cm wide, and 10 cm high. Nontoxic paint was used to paint the forepaws (red) and hindpaws (blue) of the mice; the mice then were placed at one end of the runway and allowed to run to a covered goal box at the other end, leaving painted footprints on the sheet of paper along the way. Each mouse was given three training sessions without painted feet and then one trial session with painted feet. The following gait parameters were then analyzed (1) stride length, the average distance between successive footprints on each side, that is, the distance between the first left forepaw and second left forepaw, etc.; or between the first left hindpaw and second left hindpaw, etc (2) stance length, the average distance between the left and right forepaws or hindpaws; (3) overlap between forepaw and hindpaw placement, the average distance between the center of the forepaw and hindpaw prints on each side, which measures foot placement accuracy and step pattern uniformity; (4) paw base, the length or forepaws or hindpaws.

### Data analysis and statistics

Electrophysiological data were analysed using Clampfit 10.7 (Molecular Devices) or Igor Pro 7.08 (WaveMetrics, Lake Oswego, OR). Statistical analysis was done using GraphPad Prism software.

### Reporting summary

Further information on research design is available in the Nature Portfolio Reporting Summary linked to this article.

## Data availability

Source data are provided within this paper. For Figure S1, data are available at http://dropviz.org and https://www.ncbi.nlm.nih.gov/geo/query/acc.cgi?acc = GSE160471. Source data are provided with this paper.

## Code availability

The Matlab script developed by Dr. Mu Zhou and used in this paper to analyze mouse tremor is available at https://github.com/behavioranalysis/Tremor-analysis.

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

## Acknowledgements

We thank Dr. K. Liakath-Ali for technical support on the Ribotag experiment, for preparing the samples used in Fig. 1b-c, Dr. M. Zhou for developing the Matlab script used in this paper to perform the tremor analysis in Fig. 2c (see ref. 44), and E. Schonfeld for analyzing the DCN RNA sequencing data in figure S1b. Schematic in Fig. 5a was created with BioRender. This work was supported by a grant from the NIMH (MH052804 to T.C.S.).

## Author contributions

A.S. and T.C.S. conceived the study. A.S. performed the experiments and analyzed the data. A.S. and T.C.S. wrote the manuscript. All authors contributed to the article and approved the submitted version.

## Competing interests

The authors declare no competing interests.
