## [Peer Review File · Nature Communications]

REVIEWER COMMENTS

Reviewer #1 (Remarks to the Author):

The manuscript by Scip and Sudhof assesses roles of neuexins and LAR-RPTPs in synapse assembly and of their redundancy in the process. It combines mouse genetics with morphological, behavioral and electrophysiological readouts to establish that these proteins have important redundant roles in the formation of PC->DCN synapses.

Strengths:

Overall, the approach builds on massive, rigorous genetic experiments to establish roles for LAR-RPTPs and neuexin proteins. A multitude of readouts is used to build the main point; these approaches include analyses of motor behaviors, slice electrophysiology and morphology. The readouts complement each other and there is no doubt left that there is a robust effect on synapse formation when these proteins are removed. The problem that the study addresses is a fundamental question in cellular neuroscience that has been difficult to tackle. While many of the synaptic cell adhesion proteins are synaptogenic in vitro, loss of function approaches have often failed to convincingly establish roles in synapse formation. As such, the massive mouse genetic groundwork pursued here is both necessary and laudable to address this fundamental question. The manuscript is well written, the data are rigorously acquired, and the discussion is balanced and points out weaknesses with clarity. In my view, the paper is suitable for Nature Communications with only minor revisions.

Additional comments:

One point that should be addressed with additional data analyses and/or a new experiment is whether overall PC axon growth is affected by simultaneous deletion of neuexins and LAR-RPTPs. Many experiments show that there are fewer synapses in DCN. It is unclear whether PC axons grow and branch normally but make fewer synapses, or whether there is an effect on axon branching and a less developed axon leads to a lower number of synapses upon the strong genetic manipulation. This point should also be added to the discussion. Along those lines, in fig. 3, analyses of the number of calbindin-positive PCs, and their size and staining intensity and appearance seems necessary. At least in one sample image (last image in panel 3f), it appears that calbindin staining intensity is affected by deletion of the six genes.

Minor points:

- In fig. 1, the number of clusters is given, but not the number of slices and mice analyzed. This information should be provided. There is also a mismatch in panel labeling between figure and legend.
- In fig.1, PTPRs antibodies are used for STORM microscopy; antibody validation refers to a previous paper, but as far as I can tell, the antibodies are only validated for Westerns (with a number of cross-reactive bands at low molecular weight). While antibody validation may be difficult in the given experiments (conditional knockout in a small set of neurons), at least a word of caution should be added. The overall conclusion of synapse formation does not hinge upon this validation.
- Statistics: throughout the manuscript, t-tests were used for pairwise comparisons. It is unclear whether the data follow the assumptions of these tests (variance, distribution). This should be tested, and in some cases non-parametric tests might be more appropriate. In any case, the basis for the choice of the statistical tests should be described in the statistics section.

Reviewer #2 (Remarks to the Author):

The study by Sclip and Sudhof examined the functional redundancy of neurexins and LAR-PTPRs in synapse formation, specifically at PC Δ DCN synapses. The authors used single-cell RNAseq data and riboTag profiling to identify the expression of neurexins and LAR-PTPRs in Purkinje cells and deep cerebellar nuclei. They also conducted dSTORM analysis to show that neurexins and LAR-PTPRs co-localized in synapses and are part of the same nanoclusters. They used conditional KO mice to investigate the effect of neurexin and/or LAR-PTPR deletions on PC to DCN synapses. They found that PV-Cre Nrnx123-PtprDFS 6cKO mice exhibited major motor behavior abnormalities, indicative of a cerebellar phenotype. This phenotype was more exacerbated in the Nrnx123-PtprDFS 6cKO than in the neurexin123-triple KO or the LAR-PTPR triple KO. Further, synapse formation, mIPSC frequency and evoked IPSC were greatly reduced in the Nrnx123-PtprDFS 6cKO than in either neurexin123-triple KO or LAR-PTPR triple KO.

The study presents the notion that all Nrnxns and all Ptpers are redundant. However, this proposition may not be entirely accurate. The authors themselves have shown that Nrnx2 has effects on synapse function opposing that of Nrnx1 and Nrnx2, particularly with regards to mEPSC frequency and synapse numbers. Thus, it remains unclear what the triple neurexin KO establishes in terms of advancement of the field. Additionally, the authors' findings demonstrate that neurexin 2b, 3a and PtpoS are the predominant organizers expressed in PC, making it unclear why the authors did not specifically generate a knockout of these three variants rather than proceeding with the Nrnx123-PtprDFS 6cKO. Regrettably, the study's contribution to elucidating the roles of synapse organizers remains questionable, as it fails to expound upon the specific functions of neurexin 2b, 3a, and PtpoS in the PC to DCN synapses. Additionally, the notion of neurexin and PtpoS collaboration is hardly novel, having been previously demonstrated in a seminal study by Roppongi and colleagues (Neuron 2020), yet curiously left unacknowledged in the present work. The study reveals that neurexins and Ptpers operate within akin pathways, a rather unsurprising finding. Nevertheless, it remains crucial for the authors to meticulously explicate the individual roles of neurexin 2b, 3a and PtpoS in PC to DCN synapses. Equally imperative is the need to uphold scientific rigor by appropriately citing all relevant and applicable studies, which enhances the integrity of the study and contributes to the advancement of the field.

Reviewer #3 (Remarks to the Author):

This manuscript describes the effects in mice of a sextuple knockout of all the neurexins and LAR-type receptor phosphotyrosine phosphatases (LAR-PTPRs). These proteins are synaptic adhesion molecules that have been shown to play roles in synapse formation in reduced systems, but which have apparently redundant or mutually compensating roles at real synapses. The goal of the present study is to test the consequence of the loss of all six members of the two families of proteins. The results provide evidence that loss of LAR-PTPRs alone leads only to modest changes in synapses; loss of neurexins produces some reduction in synapse formation and associated deficits; and loss of both groups of proteins has severe effects. The data thus provide evidence for a functional redundancy even between these very different proteins. The work on the whole is convincing although there are a few perplexing points, detailed below.

1. The presentation is a bit superficial in parts, and the authors could be more skeptical and critical, if only for reasons of formal logic, of the results as they unfold, in part by acknowledging more of the complexities within the work than they do. For instance, the manuscript is written as though Purkinje cells are the only neurons affected by the mutations, yet the deletions are in *all* parvalbumin expressing cells. These include Purkinje cells, but also molecular layer interneurons in the cerebellum, many inhibitory neurons in the cortex (including motor cortex), basal ganglia, spinal cord, and even muscles. Yet this wide expression is never mentioned, and all the behavioral data are interpreted as though they can be attributed exclusively to cerebellar (Purkinje) deficits (line 116: "major motor behavior abnormalities, indicative of a cerebellar phenotype" and related

places). This should be handled more clearly, if only for the sake of not misleading readers. It should be straightforward to write clearly about it, especially since it really doesn't matter to the primary line of conclusions whether the deficits are really cerebellar or not.

2. Figure 3f. The vGluT2 staining is not very clear. Perhaps a low-mag image that shows the physical extent of climbing fiber innervation would be more informative, and would better address the basic question of whether there are gross morphological differences.

3. Figure 4, Some assessment of the significance/limitations of staining intensity, area, and cluster density would be informative. While there is no question that there are reductions evident in the sextuple KO, the reader is left trying to understand things like the "right-shift" in the LAR-PTPR KO, which hints at increases (do we care? why?) and the high variance in some measures but not others.

4. The electrophysiology confirms the expectations from the loss in synapses. Nevertheless, the methods are surprising. It is impressive that the authors got recordings from cerebellar slices from post-weanling mouse brains cut ice cold (which usually shatters the myelin and damages the DCN cells), and with chloride fills (which DCN cells do not always tolerate well). These must have been heroic experiments. In which nucleus/nuclei were the DCN cells recorded? What was the holding current and access resistance? Some more information about recordings and exclusion/inclusion criteria should be given.

Minor points

1. Intro p.3 line 39, please spell out LAR the first time it is presented.

2. The writing should be examined for accuracy in several places. Two are noted here:

a. For instance, the authors conflate "Purkinje cells" and "the cerebellum" as well as "granule cells" in a few places. In Supp. Fig 1c, it looks like Nr1n1 is in granule cells rather than Purkinje cells although it is hard to tell. If the figure is meant to indicate Purkinje cell expression, it should be shown at higher gain. If Supp Fig 1a is sufficient to make the case, the Nr1n1 image from 1c could be removed.

b. Similarly, lines 91-92 say "The cerebellum contains the majority of neurons in brain." This is true but this refers to the granule cells which is not a topic of their study. Then it says "PC \leftrightarrow DCN synapses are the only output pathways for cerebellar neurons" which is a conflation of the fact that PCs are the sole output of the cerebellar cortex, and DCN neurons are the output of the cerebellum. The synapses between these cells are in the core of the cerebellum and have nothing to do with output.

3. Figure 1e and 1f legend are mismatched with the panels (should be 1f and 1g)

4. Figure 1g, left panel y-axis. Please define "localization count." Is this simply the number of puncta?

5. Figure 3b etc. please indicate what the numbers on the plots represent (18/3, 19/3) presumably number of some kind of measurement in some number of mice?

6. Discussion 309-312, check sentence. "Despite the effect size...is very large..." Should "Despite" be "Although" (?)

REVIEWER COMMENTS and CHANGES INSTITUTED INTO THE REVISED MANUSCRIPT for Alessandra Scipio entitled "..."

We thank the reviewers for their constructive comments, which we found very helpful in revising the manuscript. As described below, we have made major changes in the paper and added significant additional experimental data in response to the reviewers' comments. In the following, we cite the reviewers' comments in full in *italic* typeface and provide our responses in dark blue typeface.

Reviewer #1:

The manuscript by Scipio and Sudhof assesses roles of neurexins and LAR-RPTPs in synapse assembly and of their redundancy in the process. It combines mouse genetics with morphological, behavioral and electrophysiological readouts to establish that these proteins have important redundant roles in the formation of PC->DCN synapses.

Strengths:

Overall, the approach builds on massive, rigorous genetic experiments to establish roles for LAR-RPTPs and neurexin proteins. A multitude of readouts is used to build the main point; these approaches include analyses of motor behaviors, slice electrophysiology and morphology. The readouts complement each other and there is no doubt left that there is a robust effect on synapse formation when these proteins are removed. The problem that the study addresses is a fundamental question in cellular neuroscience that has been difficult to tackle. While many of the synaptic cell adhesion proteins are synaptogenic in vitro, loss of function approaches have often failed to convincingly establish roles in synapse formation. As such, the massive mouse genetic groundwork pursued here is both necessary and laudable to address this fundamental question. The manuscript is well written, the data are rigorously acquired, and the discussion is balanced and points out weaknesses with clarity. In my view, the paper is suitable for Nature Communications with only minor revisions.

We appreciate the reviewers positive and balanced comments – it is truly gratifying to receive an understanding of the challenges we have faced in this project

Additional comments:

One point that should be addressed with additional data analyses and/or a new experiment is whether overall PC axon growth is affected by simultaneous deletion of neurexins and LAR-RPTPs. Many experiments show that there are fewer synapses in DCN. It is unclear whether PC axons grow and branch normally but make fewer synapses, or whether there is an effect on axon branching and a less developed axon leads to a lower number of synapses upon the strong genetic manipulation. This point should also be added to the discussion.

To address this point, we have measured the extension of Purkinje cell axons into the deep cerebellar nuclei (new Figure S5) and additionally amended the Discussion as recommended. In this new experiment we performed injections of AAV-CAG-GFP viruses into the PC layer of the cerebellum to anterogradely trace Purkinje cell axons in Nrnx123-

PtprDFS 6cKO mice. Using this method, we were able to visualize Purkinje cells and to trace their axons. We measured the fraction of Purkinje cell axons area in the DCN and normalized it to the number of GFP+ PCs to correct for differences in injection sites. Our data show that deletion of Nrnx and LAR-Ptprs does not seem to affect axonal growth.

Along those lines, in fig. 3, analyses of the number of calbindin-positive PCs, and their size and staining intensity and appearance seems necessary. At least in one sample image (last image in panel 3f), it appears that calbindin staining intensity is affected by deletion of the six genes.

We have now performed a detailed quantification of the number of calbindin-positive Purkinje cells (expressed as density), the size of their soma, and their calbindin staining intensity, as suggested by the reviewer (new Figure S4).

Minor points:

- In fig. 1, the number of clusters is given, but not the number of slices and mice analyzed. This information should be provided. There is also a mismatch in panel labeling between figure and legend.

We have added in the figure legend detailed information on the number of ROIs/mice analyzed. Also, we thank the reviewer for catching the mistake in the figure legend panels that we have now corrected.

- In fig.1, PTPRs antibodies are used for STORM microscopy; antibody validation refers to a previous paper, but as far as I can tell, the antibodies are only validated for Westerns (with a number of cross-reactive bands at low molecular weight). While antibody validation may be difficult in the given experiments (conditional knockout in a small set of neurons), at least a word of caution should be added. The overall conclusion of synapse formation does not hinge upon this validation.

We have added a new experiment documented in an additional supplementary figure (Figure S2) to address this point. In this figure we show representative images of the DCN of PV-negative and PV-positive Nrnx123-PtprDFS 6cKO mice stained with the PAN-NRXN and PTPRS antibodies. This figure shows that NRXN- and PTPRS-staining is massively reduced in slices from PV-Cre mice, with the remaining puncta likely due to excitatory inputs into the DCN from neurons that do not express PV-Cre.

- Statistics: throughout the manuscript, t-tests were used for pairwise comparisons. It is unclear whether the data follow the assumptions of these tests (variance, distribution). This should be tested, and in some cases non-parametric tests might be more appropriate. In any case, the basis for the choice of the statistical tests should be described in the statistics section.

We now show statistical significances obtained with Mann-Whitney test for all pairwise comparison.

Reviewer #2:

*The study by Sclijp and Sudhof examined the functional redundancy of neurexins and LAR-PTPRs in synapse formation, specifically at PC \leftrightarrow DCN synapses. The authors used single-cell RNAseq data and riboTag profiling to identify the expression of neurexins and LAR-PTPRs in Purkinje cells and deep cerebellar nuclei. They also conducted dSTORM analysis to show that neurexins and LAR-PTPRs co-localized in synapses and are part of the same nanoclusters. They used conditional KO mice to investigate the effect of neurexin and/or LAR-PTPR deletions on PC to DCN synapses. They found that PV-Cre *Nrxn123-PtprDFS* 6cKO mice exhibited major motor behavior abnormalities, indicative of a cerebellar phenotype. This phenotype was more exacerbated in the *Nrxn123-PtprDFS* 6cKO than in the neurexin123-triple KO or the LAR-PTPR triple KO. Further, synapse formation, mEPSC frequency and evoked IPSC were greatly reduced in the *Nrxn123-PtprDFS* 6cKO than in either neurexin123-triple KO or LAR-PTPR triple KO.*

*The study presents the notion that all *Nrxns* and all *Ptprs* are redundant. However, this proposition may not be entirely accurate. The authors themselves have shown that *Nrxn2* has effects on synapse function opposing that of *Nrxn1* and *Nrxn2*, particularly with regards to mEPSC frequency and synapse numbers. Thus, it remains unclear what the triple neurexin KO establishes in terms of advancement of the field.*

We believe the reviewer misunderstood our data and text. Nowhere do we present the notion that “*all *Nrxns* and all *Ptprs* are redundant*”. Instead, as summarized nicely by Reviewer #1, the paper uses a conditional deletion of all *Nrxns* and *Ptprs* to probe their overall role, without an attempt to deconstruct the functions and contributions of individual genes. Such an analysis will require a large amount of additional work from many laboratories and is not the goal of the current paper. To ask the question we are posing in the present study, the triple *Nrxn* and *Ptprs* KOs and the sextuple *Nrxn-Ptpr* KOs are essential. No previous study has probed this question before.

*Additionally, the authors' findings demonstrate that neurexin 2b, 3a and *PtprS* are the predominant organizers expressed in PC, making it unclear why the authors did not specifically generate a knockout of these three variants rather than proceeding with the *Nrxn123-PtprDFS* 6cKO.*

Again, the reviewer appears to misunderstand the data we present. Our findings do NOT “*demonstrate that neurexin 2b, 3a and *PtprS* are the predominant organizers expressed in PC*”. Instead, our data only show that these isoforms are the most abundantly expressed isoforms that are enriched in Purkinje cells. RNAseq data only provide initial insights into biological relevance since isoforms that are expressed at lower abundance – not even at low levels, just less than the predominantly expressed isoforms – are often functionally decisive but are not detected in many RNAseq datasets.

Regrettably, the study's contribution to elucidating the roles of synapse organizers remains questionable, as it fails to expound upon the specific functions of neurexin 2b, 3a, and PtpoS in the PC to DCN synapses.

This is the same criticism as above. Just targeting a subset of Nrns and Ptps would defeat the purpose of the study.

Additionally, the notion of neurexin and PtpoS collaboration is hardly novel, having been previously demonstrated in a seminal study by Roppongi and colleagues (Neuron 2020), yet curiously left unacknowledged in the present work. The study reveals that neurexins and Ptps operate within akin pathways, a rather unsurprising finding.

The excellent Roppongi et al. paper from the Siddiqui lab is a nice study that contains interesting data but few functional experiments on either neurexins or Ptp's. Surely the reviewer won't claim that this paper addresses the question we pose. After all, this paper studies LRRTM4 and presents beautiful biochemical data but has little relevance to our study apart from reporting in vitro interactions.

Nevertheless, it remains crucial for the authors to meticulously explicate the individual roles of neurexin 2b, 3a and PtpoS in PC to DCN synapses.

This is again the same criticism as above. As explained above, just targeting a subset of Nrns and Ptps would defeat the purpose of the study.

Equally imperative is the need to uphold scientific rigor by appropriately citing all relevant and applicable studies, which enhances the integrity of the study and contributes to the advancement of the field.

We apologize if we overlooked a relevant paper and would appreciate guidance on what studies the reviewer believes besides the Roppongi/Siddiqui paper that we felt was not directly related to our current work.

Reviewer #3:

This manuscript describes the effects in mice of a sextuple knockout of all the neurexins and LAR-type receptor phosphotyrosine phosphatases (LAR-PTPRs). These proteins are synaptic adhesion molecules that have been shown to play roles in synapse formation in reduced systems, but which have apparently redundant or mutually compensating roles at real synapses. The goal of the present study is to test the consequence of the loss of all six members of the two families of proteins. The results provide evidence that loss of LAR-PTPRs alone leads only to modest changes in synapses; loss of neurexins produces some reduction in synapse formation and associated deficits; and loss of both groups of proteins has severe effects. The data thus provide evidence for a functional redundancy even between these very different proteins. The work on the whole is convincing although there are a few perplexing

points, detailed below.

1. The presentation is a bit superficial in parts, and the authors could be more skeptical and critical, if only for reasons of formal logic, of the results as they unfold, in part by acknowledging more of the complexities within the work than they do. For instance, the manuscript is written as though Purkinje cells are the only neurons affected by the mutations, yet the deletions are in **all** parvalbumin expressing cells. These include Purkinje cells, but also molecular layer interneurons in the cerebellum, many inhibitory neurons in the cortex (including motor cortex), basal ganglia, spinal cord, and even muscles. Yet this wide expression is never mentioned, and all the behavioral data are interpreted as though they can be attributed exclusively to cerebellar (Purkinje) deficits (line 116: “major motor behavior abnormalities, indicative of a cerebellar phenotype” and related places). This should be handled more clearly, if only for the sake of not misleading readers. It should be straightforward to write clearly about it, especially since it really doesn’t matter to the primary line of conclusions whether the deficits are really cerebellar or not.

We agree and have revised the manuscript accordingly.

2. Figure 3f. The vGluT2 staining is not very clear. Perhaps a low-mag image that shows the physical extent of climbing fiber innervation would be more informative, and would better address the basic question of whether there are gross morphological differences.

We have added additional representative images in Figure S4 to render the vGluT2 staining more clear.

3. Figure 4, Some assessment of the significance/limitations of staining intensity, area, and cluster density would be informative. While there is no question that there are reductions evident in the sextuple KO, the reader is left trying to understand things like the “right-shift” in the LAR-PTPR KO, which hints at increases (do we care? why?) and the high variance in some measures but not others.

Following the reviewer’s comments, we have amended the text to improve the clarity of the information provided.

For staining experiments, we used different parameters to assess potential phenotypes. Staining intensity is a common measurement which correlates with a protein’s abundance. This measure can vary as a function of changes in protein expression, but also is dependent on variables in the sample preparation (i.e. quality of perfusion, staining protocol) and obviously the imaging parameters (which are thus held constant). To account for this technical variability, intensity of a signal can be normalized to an internal signal (for example in Figure 3e, h we normalized the intensity of vGluT1 and vGluT2 staining to calbindin).

For staining in which individual synaptic puncta can be resolved (vGluT2 in figure 3f, g), it is standard practice to measure the cluster density (number of puncta/ROI area). As an alternative, for staining where it is harder to resolve individual puncta (see vGAT staining

in Figure 4a), it is possible to analyze the area covered by the signal in a defined ROI. These measurements correlate with both the abundance of that specific synaptic marker and the number of synapses.

Regarding the increase in vGAT staining and intensity observed in Figure 4b, it is important to note that these changes are small and do not reach significance when true biological replicates (number of mice, see summary plots) are plotted, as opposed to pseudoreplicates (number of ROIs), which increase statistical power. However, it remains a possibility that deletion of LAR-PTPRs leads to small changes in the number/strength of PC→DCN synapses. Similar trends are also observed in the mIPSC amplitude analysis (Figure 6b3) and in the rotarod experiments (see learning rate in Figure 2i4). However, the effect size of these trends is modest and statistical significance is not reached.

4. The electrophysiology confirms the expectations from the loss in synapses. Nevertheless, the methods are surprising. It is impressive that the authors got recordings from cerebellar slices from post-weanling mouse brains cut ice cold (which usually shatters the myelin and damages the DCN cells), and with chloride fills (which DCN cells do not always tolerate well). These must have been heroic experiments. In which nucleus/nuclei were the DCN cells recorded? What was the holding current and access resistance? Some more information about recordings and exclusion/inclusion criteria should be given.

We appreciate the reviewer's acknowledgement of the difficulties of these experiments and have added further details of these recordings as well as data on the passive electrical properties that we measured (new Figure S7).

Cells in all three deep cerebellar nuclei were included in the analysis. We believe this choice is valid, given the overlap in the transcriptional profile of neurons in the medial, interposed, and lateral deep cerebellar nuclei (see Figure S1b).

For the inclusion/exclusion parameters, all stable cells that had a Ra lower than 20 mOhm were recorded and included in the analysis. Quality of the recordings was assessed during experiments. All experiments were performed with anonymized samples, such that the experimenter was unaware of the genotype of a sample when performing recordings and analyses.

Minor points

1. Intro p.3 line 39, please spell out LAR the first time it is presented.

This has been corrected

*2. The writing should be examined for accuracy in several places. Two are noted here:
a. For instance, the authors conflate "Purkinje cells" and "the cerebellum" as well as "granule cells" in a few places. In Supp. Fig 1c, it looks like Nrnx1 is in granule cells rather than Purkinje cells although it is hard to tell. If the figure is meant to indicate Purkinje cell expression, it should be shown at higher gain. If Supp Fig 1a is sufficient to make the case, the Nrnx1 image from 1c could be removed.*

We agree with the reviewer and have removed Figure S1c. We have also revised the text to be more precise in the wording as recommended.

b. Similarly, lines 91-92 say “The cerebellum contains the majority of neurons in brain.” This is true but this refers to the granule cells which is not a topic of their study. Then it says “PC↔DCN synapses are the only output pathways for cerebellar neurons” which is a conflation of the fact that PCs are the sole output of the cerebellar cortex, and DCN neurons are the output of the cerebellum. The synapses between these cells are in the core of the cerebellum and have nothing to do with output.

Again, we have revised the text to be more precise.

3. Figure 1e and 1f legend are mismatched with the panels (should be 1f and 1g)

Agreed – this has been corrected

4. Figure 1g, left panel y-axis. Please define “localization count.” Is this simply the number of puncta?

The localization count is the count of blinking events occurring within a cluster (puncta) during the acquisition time.

5. Figure 3b etc. please indicate what the numbers on the plots represent (18/3, 19/3) presumably number of some kind of measurement in some number of mice?

Agreed – we now specify that these are the number of slices/mice

6. Discussion 309-312, check sentence. “Despite the effect size...is very large....” Should “Despite” be “Although” (?)

Agreed – this has been corrected

We appreciate the reviewers' comments and hope that the revised manuscript with the additional data can be deemed acceptable for publication.

REVIEWERS' COMMENTS

Reviewer #1 (Remarks to the Author):

The authors have addressed my comments and I fully support publication of the manuscript in its current form in Nature Communications.

Reviewer #2 (Remarks to the Author):

The authors have addressed my concerns satisfactorily.

Reviewer #3 (Remarks to the Author):

The authors have addressed most of my comments. Minor remaining points:

1. Lines 97-99 "As the only output pathway for cerebellar neurons, PC->DCN synapses are thus placed in a central position in cerebellar circuits (35-37)."

These sentences still sound like the PC->DCN synapse is itself an output. Perhaps edit to something like: "The DCN forms the only output pathway of the cerebellum; PC->DCN synapses are thus placed in a central position in cerebellar circuits (35-37)."

2. The electrophysiological parameters are still incomplete in the manuscript, although Figure S7 is helpful.

a. The response indicates: "For the inclusion/exclusion parameters, all stable cells that had a R_a lower than 20 mOhm were recorded and included in the analysis." This useful information should be in the manuscript. It looks like one or two cells in the plot in S7A have an access resistance >20 MOhm, though; perhaps edit the value accordingly and state in methods.

b. The response also indicates, "Quality of the recordings was assessed during experiments." This is expected, but some statement about what parameters were assessed to establish quality should be included in the manuscript, e.g., stability of R_a , R_{input} , or a lack of change in leak, etc.

Reviewer #1 (Remarks to the Author):

The authors have addressed my comments and I fully support publication of the manuscript in its current form in Nature Communications.

Reviewer #2 (Remarks to the Author):

The authors have addressed my concerns satisfactorily.

Reviewer #3 (Remarks to the Author):

The authors have addressed most of my comments. Minor remaining points:

1. Lines 97-99 “As the only output pathway for cerebellar neurons, PC->DCN synapses are thus placed in a central position in cerebellar circuits (35-37).”

These sentences still sound like the PC->DCN synapse is itself an output. Perhaps edit to something like: “The DCN forms the only output pathway of the cerebellum; PC->DCN synapses are thus placed in a central position in cerebellar circuits (35-37).”

2. The electrophysiological parameters are still incomplete in the manuscript, although Figure S7 is helpful.

a. The response indicates: “For the inclusion/exclusion parameters, all stable cells that had a R_a lower than 20 mOhm were recorded and included in the analysis.” This useful information should be in the manuscript. It looks like one or two cells in the plot in S7A have an access resistance >20 MOhm, though; perhaps edit the value accordingly and state in methods.

b. The response also indicates, “Quality of the recordings was assessed during experiments.” This is expected, but some statement about what parameters were assessed to establish quality should be included in the manuscript, e.g., stability of R_a , R_{in} , or a lack of change in leak, etc.

We thank the reviewers for their constructive comments during the initial revision of the manuscript.

Please find here a response to reviewer 3.

The authors have addressed most of my comments. Minor remaining points:

1. Lines 97-99 “As the only output pathway for cerebellar neurons, PC->DCN synapses are thus placed in a central position in cerebellar circuits (35-37).”

These sentences still sound like the PC->DCN synapse is itself an output. Perhaps edit

to something like: “The DCN forms the only output pathway of the cerebellum; PC->DCN synapses are thus placed in a central position in cerebellar circuits (35-37).”
We thank the reviewer and have now change this sentence as suggested.

2. The electrophysiological parameters are still incomplete in the manuscript, although Figure S7 is helpful.

a. The response indicates: “For the inclusion/exclusion parameters, all stable cells that had a R_a lower than 20 mOhm were recorded and included in the analysis.” This useful information should be in the manuscript. It looks like one or two cells in the plot in S7A have an access resistance >20 MOhm, though; perhaps edit the value accordingly and state in methods.

We added this information in the manuscript and have corrected the value according to the reviewer suggestion.

b. The response also indicates, “Quality of the recordings was assessed during experiments.” This is expected, but some statement about what parameters were assessed to establish quality should be included in the manuscript, e.g., stability of R_a , R_{input} , or a lack of change in leak, etc.

We added the following statement in the method section as suggested by the reviewer: “Quality of the recordings was assessed during the experiments. Cells with unstable leak current, R_a , and R_{in} were not recorded or excluded from the analysis.”